# A ribosomally synthesised and post-translationally modified peptide containing a β-enamino acid and a macrocyclic motif

Shan Wang [1,8], Sixing Lin[2,8], Qing Fang [1], Roland Gyampoh[3], Zhou Lu[1], Yingli Gao[1,4], David J. Clarke [5], Kewen Wu[1], Laurent Trembleau [1], Yi Yu [2] ✉, Kwaku Kyeremeh [3] ✉, Bruce F. Milne [1,6] ✉, Jioji Tabudravu [7] ✉ & Hai Deng [1] ✉

Ribosomally synthesized and post-translationally modified peptides (RiPPs) are structurally complex natural products with diverse bioactivities. Here we report discovery of a RiPP, kintamdin, for which the structure is determined through spectroscopy, spectrometry and genomic analysis to feature a *bis*-thioether macrocyclic ring and a β-enamino acid residue. Biosynthetic investigation demonstrated that its pathway relies on four dedicated proteins: phospho-transferase KinD, Lyase KinC, kinase homolog KinH and flavoprotein KinI, which share low homologues to enzymes known in other RiPP biosynthesis. During the posttranslational modifications, KinCD is responsible for the formation of the characteristic dehydroamino acid residues including the β-enamino acid residue, followed by oxidative decarboxylation on the C-terminal Cys and subsequent cyclization to provide the *bis*-thioether ring moiety mediated by coordinated action of KinH and KinI. Finally, conserved genomic investigation allows further identification of two kintamdin-like peptides among the *kin*-like BGCs, suggesting the occurrence of RiPPs from actinobacteria.

Cyclopeptides are a subset of secondary metabolites that can be of ribosomal or non-ribosomal origin. While non-ribosomal peptides (NRPs) have long been studied, the chemical diversity of peptides from ribosomal origin, now called ribosomally synthesised and post-translationally modified peptides (RiPPs), have only been recognised relatively recently and across all three domains of life during the last decade, as a result of recent advances in genome sequencing technology[1]. It is now known that the majority of RiPPs are synthesized from precursor peptides, typically ranging from 20 to 110 amino acid residues in length, encoded by a structural gene[2].

Although the RiPPs exploit only the 20 proteinogenic amino acids, extensive post-translational modifications (PTMs) increase their structural diversity. Dehydroalanine (Dha) and (Z)-dehydrobutyrine ((Z)-Dhb) residues are commonly found in the RiPPs. Dha and (Z)-Dhb residues in RiPPs are derived from serine/cysteine and threonine respectively[3]. In some cases, D-amino acid residues, such as D-Ala, are introduced in lanthipeptides by an $F_{420}H_2$-dependent reductase[4]. The structural complexity of RiPPs extends to a range of cyclic motifs, including noncanonical thioether amino acid residues, such as lanthionine (Lan), labionin (Lab)[5], 2-aminovinly-cysteine (AviCys), and

[1]Department of Chemistry, University of Aberdeen, Aberdeen AB24 3UE, UK. [2]Key Laboratory of Combinatorial Biosynthesis and Drug Discovery (MOE) and Hubei Province Engineering and Technology Research Centre for Fluorinated Pharmaceuticals, School of Pharmaceutical Sciences, Wuhan University, Wuhan 430071, China. [3]Department of Chemistry, University of Ghana, P.O. Box LG56 Legon-Accra, Ghana. [4]College of Marine Life and Fisheries, Jiangsu Ocean University, Lianyungang, Jiangsu Province, China. [5]EastChem, School of Chemistry, University of Edinburgh, Edinburgh EH9 3FJ, UK. [6]CFisUC, Department of Physics, University of Coimbra, Rua Larga 3004-516 Coimbra, Portugal. [7]School of Natural Sciences, University of Central Lancashire, PR1 2HE Preston, England, United Kingdom. [8]These authors contributed equally: Shan Wang, Sixing Lin. ✉e-mail: yu_yi@whu.edu.cn; kkyeremeh@ug.edu.gh; bfmilne@uc.pt; JTabudravu@uclan.ac.uk; h.deng@abdn.ac.uk

AviCys-labionin (Avionin, a decarboxylated analogue of Lab)[6] (Fig. 1a), which are enzymatically installed post-translationally. Among these cyclic residues, Lan residues are well-understood in the biosynthesis of lanthipeptides (Fig. 1b). The enzymes responsible for the Lan moiety catalyse the dehydration of either Ser or Thr residues to generate Dha or Dhb, respectively, followed by the intramolecular Michael-like addition of the side chain of Cys to Dha or Dhb to form the Lan or methyl-Lan motif, respectively. When a second Dha motif is involved in the class III lanthipeptide NAI-112 1 (Fig. 1a), the triamino triacid Lab residues can be formed (Fig. 1b)[5]. However, the factors that determine Lan or Lab formation are not yet understood[2].

Both AviCys and Avionin residues are found in several RiPPs including lanthipeptides, thioamitides, lipolanthines, lanthidins and linaridins (Fig. 1a)[2]. In the case of AviCys residues in lanthipeptides, the PTMs normally involves the dehydration of the upstream Ser or Thr residue of a precursor peptide, followed by oxidative decarboxylation of C-terminal Cys and subsequent Michael addition of the resultant thioenol nucleophile onto the preceding Dha or Dhb residue to provide AviCys or AviMeCys residue, respectively. In this pathway, the formation of the reactive thioenol species from the C-terminal Cys is catalysed by the flavoprotein LanD belonging to the family of flavin-dependent cysteine decarboxylases (HFCDs) (Fig. 1c)[6]. However, this is

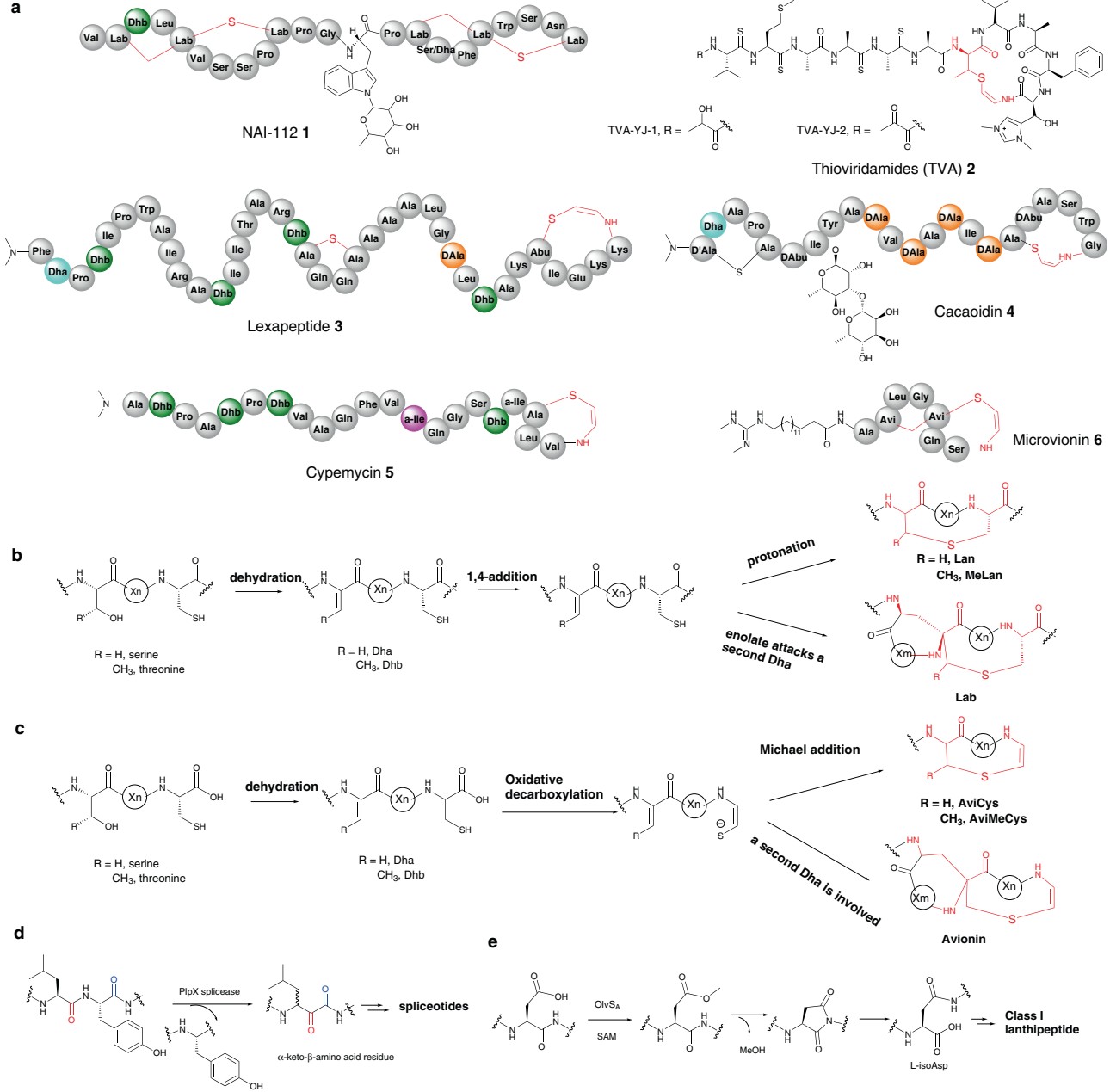

**Fig. 1 | Known thioether and β-amino acid residues-associated structures in RiPPs and proposed mechanisms. a** Selected RiPPs containing a thioether residue (red). mechanistic comparison in the formation of thioether residue Lan/Lab (**b**) and AviCys/Avionin (**c**). **b** For Lan (top) and Lab (bottom) moieties in lanthipeptides, the factors determine Lan/Lab formation are not yet understood. **c** The enzymes responsible for the formation of AviCys/AviMeCys residues are either LanD homologues in NAI-112 **1** or the coordinated action of HFCD-like TvaE$_{S-87}$ and inactive kinase homologue TvaF$_{S-87}$ in thioviridamides **2**. The enzymes responsible

for AviCys (top) and Avionin (bottom) residues in **3**, **4**, **5** and **6** have not yet been determined. **d** Example of a spliceiase reaction to provide α-keto-β-amino acid residue. Stereochemistry at the preceding Leu α-carbon has not yet been defined. **e** Example of a SAM-dependent methyltransferase OlvS$_A$ to catalyse a re-arrangement of Asp residue to L-isoAsp in Class I lanthipeptides. Other non-proteinogenic amino acid residues modified by PTMs are colour coded. Dhb (2,3-dehydrobutyrine), deep green a-Ile (allo-isoleucine), pink Dha (dehydroalanine) light blue, D-Ala (D-alanine) orange.

not the case for thioviridamide (TVA) **2** (Fig. 1a), a thioamide RiPP, in *Streptomyces sp*. NRRL S-87. Recent biochemical analysis demonstrated that the formation of AviCys residue in TVA relies on the coordination of two-component enzyme complex, the LanD-like flavoprotein TvaF[S-87] and an inactive kinase-like TvaE[S-87][7]. TvaE[S-87] and TvaF[S-87] function together by binding tightly with the leader peptide (LP) sequence of the precursor peptide to catalyse the decarboxylation of the C-terminal Cys residue, followed by subsequent addition to provide the corresponding AviCys moiety (Fig. 1b)[7]. Interestingly, while LxmX and Cao14, in the pathways of the class VI lanthipeptide lexapeptide **3**[4], and the lanthidin cacaoidin **4** (Fig. 1a)[8,9], respectively, appear to be TvaE[S-87] homologues, LxmK and CaoI are HFCD-like decarboxylases, suggesting that the AviCys moieties in these RiPPs may also result from the actions of two-component enzyme complexes. AviCys residues are also present in a subclass of lanaridin RiPPs, such as cypemycin **5** (Fig. 1a)[10]. Biochemical analyses indicated that no AviCys is formed when an assay of the recombinant HFCD-like CypD together with a synthetic core peptide (CP) mimic and necessary cofactors was performed[10], raising an unanswered question of whether an as-yet-unidentified enzyme partner is needed to coordinate the decarboxylation and cyclization. In the case of the Avionin formation in the lipolanthine RiPP microvionin **6** from the *Microbacterium arborescens* culture, the precursor peptide is post-translationally modified by oxidative decarboxylation of the C-terminal Cys and subsequent cyclization to yield avionin (Fig. 1c)[11]. Similar to Lan and Lab, the factors that determine the Avionin formation has remained to be determined.

Unlike NRPs which contain a range of β-amino acids, most RiPPs discovered so far only contains α-amino acid residues. However, recent studies indicated that β-amino acid residues can be formed in the RiPP pathways. This can be exemplified by a recent conserved genomic survey which discovered the unprecedented function of orphan radical SAM enzymes (PlpX-type spliceases) encoded in several bacterial RiPP biosynthetic gene clusters (BGCs). Biochemical analysis demonstrated that PlpX spliceases, together with its partner protein pair, PlpY, in the *plp* BGC from the cyanobacterium *Pleurocapsa* catalyses an unusual peptide backbone slicing reaction to remove all atoms of a C-terminal Tyr unit except the amide carbonyl in the precursor peptide PlpA3[12]. This generates an α-keto-β-amino residue, indicating that the corresponding mature RiPP metabolite contains β-amino acid residues (Fig. 1d). However, the structure of this RiPP associated with the *plp* BGC has not been determined[2,12]. A second example was found that L-isoaspartate, a β-amino acid residue, can be formed in Class I lanthipeptide pathways (Fig. 1e)[13]. In vitro reconstitution indicated that the *S*-adenosyl-L-methionine (SAM)-dependent *O*-methyltransferase, OlvS[A], encoded in the *olv* gene cluster from *Streptomyces olivaceus* NRRL B-3009, catalyses the re-arrangement of a highly conserved aspartate to L-isoaspartate residue (Fig. 1e)[13].

Here we report a structural and biosynthetic investigation of a macrocyclic peptide that does not fall into any of the known categories of RiPPs (Fig. 2a). It is produced by the soil isolate, *Streptomyces* sp. RK44 and bears an unprecedented chemical motif, *bis*-thioether crosslink (methyl-amino-bithionin (MAbi)), and a rare β-enamino acid ((Z)−3-amino-acrylic acid (Aaa) or Δ[Z]βAla) residues, together with motifs that are observed in other classes of RiPPs, such as D-Ala, *N*, *N*-dimethylated N-terminal Ile residues, Dha and Dhb residues (Fig. 2b). The identity of the minimal BGC (*kin*) coding the production of **7** is confirmed through heterologous expression and gene inactivation. Biosynthetic investigations indicate that the formation of dehydroamino acid, the featured β-enamino acid and MAbi bicyclic residues relies on four enzymes: the Ser/Thr kinase KinD, the lyase KinC, the kinase homologue KinH and the HFCD-like KinI, all of which share low homologies with other enzymes in known RiPP biosynthetic pathways. The enzyme partner, KinD and KinC, are responsible for the formation of all of dehydroamino acid residues in **7** in a likely

processive mode starting from the N-terminal of the CP. Subsequent oxidative decarboxylation and cyclization on the resulting dehydrated CP decorate the *bis*-thioether ring system in the coordinated action of KinI and KinH (Fig. 2c). Furthermore, site-directed mutagenesis in the CP allows the identification of key residues that play crucial roles on the PTM processes in the biosynthesis of **7**. Finally, we provide evidence of the occurrence of two kintamdin-like peptides among the *kin*-like BGCs in actinobacteria, suggesting a previously unnoticed group of RiPPs. Taken together, these results demonstrate that the recruitment of specialised classes of enzymes and CPs to biosynthetic clusters has likely increased the diversity of RiPP structures.

## Results

### Identification of kintamdin 7 from *Streptomyces* sp RK44

*Streptomyces sp*. RK44 is a recent isolate from a soil sample collected at the Kintampo waterfall in the Bono East of Ghana in 2014. Metabolite profiling of the strain revealed the presence of a high molecular weight metabolite (2507 Da) in its natural product profile[14] under laboratory culture conditions. Large-scale fermentation (10 L), followed by chemical workup allowed the isolation of pure compound **7** (3 mg).

High-resolution electrospray ionisation mass spectrometry (HR-ESIMS) analysis gave a $[M + 3H]^{3+}$ ion at *m/z* of 836.7524, indicating a neutral monoisotopic mass of 2507.2349 Da (Supplementary Fig. 1). Inspection of [1]H-NMR indicated the presence of numerous overlapping α proton signals of amino acids ($\delta_H$ 3.4–4.6 ppm), suggesting that **7** is a peptidic natural product (Supplementary Fig. 2). Product **7** displays a pair of interesting [1]H-NMR signals (5.44 and 7.20 ppm, 8.4 Hz *J*-coupling) for a peptidyl natural product, suggesting the presence of a moiety containing a *cis*-configured alkene (Fig. 2b and Supplementary Fig. 2). Interpretation of 1- and 2-D NMR spectra enabled identification of the major fragment (19 mers) of the peptide (Supplementary Figs. 3–7). Along with different proteinogenic amino acids, modified residues such as dehydroalanine (Dha) and dehydrobutyrine (Dhb) were found in **7** (Supplementary Fig. S8–11). The unit containing *cis*-alkene was identified as an unsaturated β-enamino acid, (Z)−3-amino-acrylic acid (Aaa), which to the best of our knowledge, has not been previously reported in any other RiPPs (Supplementary Figs. 12–14). The sequence tag was also confirmed through assignment of de novo analysis of $MS^n$ spectra using tandem MS fragmentation and the sequential fragment ions (Supplementary Fig. 15 and Supplementary Table 1). Almost all of the mass shifts generated can be substituted with proteinogenic amino acids except mass shifts of 69 and 83 Da which were assigned to the non-proteinogenic amino acids, Dha and Dhb. It is noteworthy that Aaa (Δ[Z]βAla), presumably a rearranged product of dehydrated serine, has the same molecular weight as Dha by MS analysis. Collectively, the NMR data combined with MS analysis allowed us to connect the major fragments of **7** in sequence although the N and C termini remained ambiguous.

### Kintamdin 7 featuring unusual chemical moieties

A genome mining strategy was then used to refine the structure of **7**. A BLAST search in the annotated RK44 genome in RAST servers[15] using the sequence tag as a probe led to the identification of a 174 bp open reading frame (*orf*) encoding the precursor peptide, KinA (Supplementary Fig. 16). With the amino acid (AA) sequence in hands, the molecular formula of **7** was established as $C_{115}H_{174}N_{28}O_{31}S_2$ based on the HR-ESIMS analysis (observed $[M + 3H]^{3+} = 836.7524$, calculated $[M + 3H]^{3+} = 836.7520$, $\Delta = 0.518$ ppm) (Supplementary Fig. 1). We then revisited the NMR spectra including the [1]H, COSY, TOCSY, HSQC, HMBC and NOESY (Supplementary Fig. 17–27 and Supplementary Table 2). An *N*, *N*-dimethyl isoleucine was found to be present at the N-terminus of **7**, a typical chemical feature of lexapeptide **3**[4], cacaoidin **4**[8,9], and linaridin family[10]. Unfortunately, fragmentation within the C-terminal structure was not observed in $MS^2$ and the $y_6$ ion was resistant to fragmentation in $MS^3$ experiments. Therefore, HRMS and

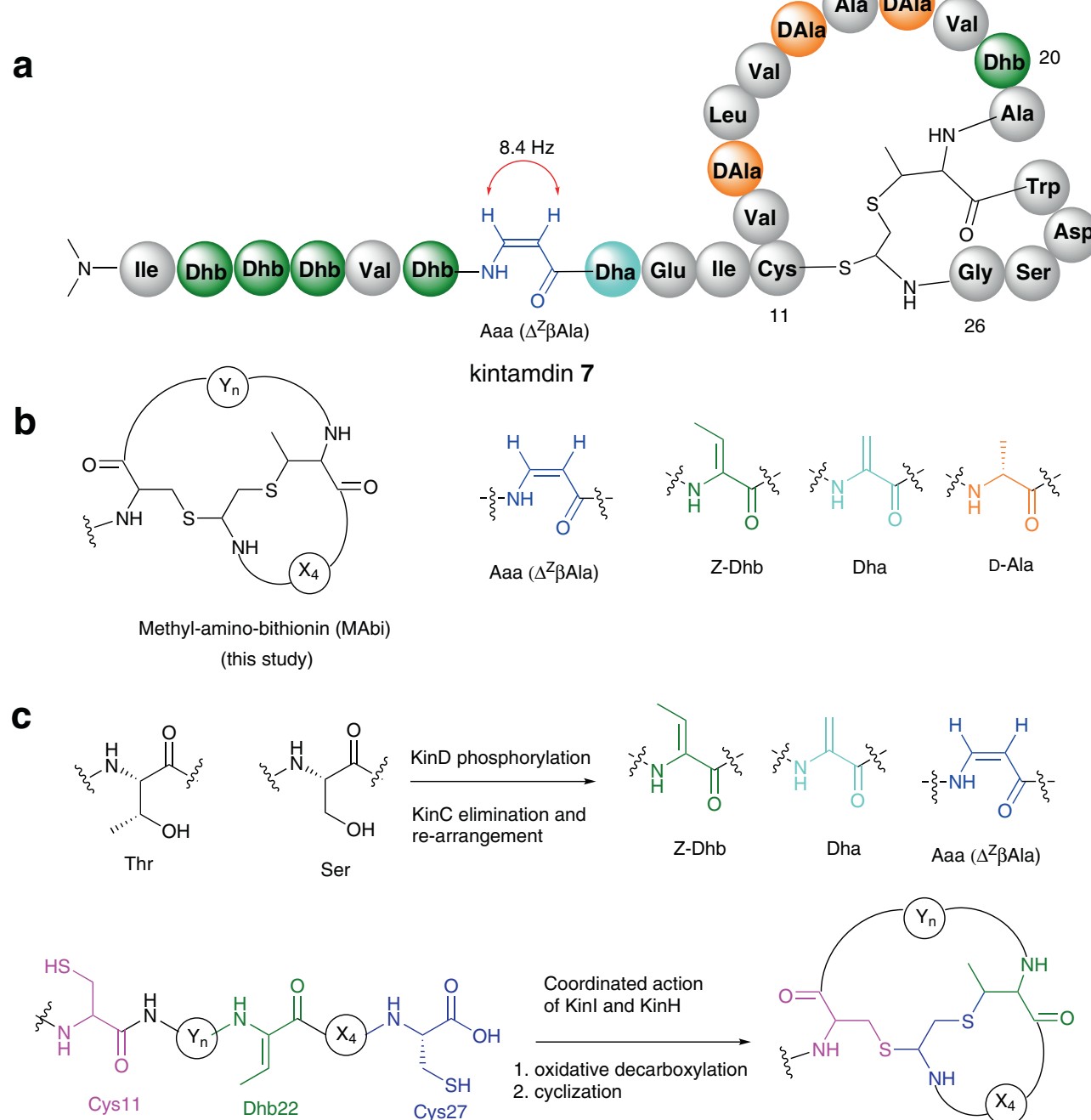

**Fig. 2 | Schematic structure and biosynthetic investigation of 7 with unique moieties highlighted in colours. a** Schematic representation of **7**. **b** Non-proteinogenic amino-acid residues resulting from different post-translational modifications (PTMs) in **7**. **c** Gene co-expression in *E. coli* demonstrated that the enzyme partner, KinD and KinC, catalyses dehydration reaction on Ser/Thr residues in the core peptide of KinA to provide dehydroamino acid residues (top), followed by a coordinated action by the HFCD-like decarboxylase KinI and the kinase homologue KinH, resulting in the formation of MAbi bicyclic residues (bottom). The individual amino acid residues involved in the MAbi were colour coded. Cys-11: pink, Cys-27: dark blue and Dhb22: green. Non-proteinogenic amino acid residues modified by PTMs are colour coded. Abbreviation: Dha dehydroalanine (light blue), Dhb dehydrobutyrine (green), D-Ala D-alanine(orange), Aaa 3-aminoacrylic acid (blue), MAbi methyl-amino-bithionin (black).

isotope fine structure analysis was used to confirm the elemental formula of the $y_6$ ion as $[C_{26}H_{34}N_7O_8S_1]^+$ (Fig. 3b–d). However, inspection of the NMR spectra together with the sequence of the precursor peptide allowed us to locate an unprecedented ring system composed of a *bis*-thioether crosslink, MAbi, at the C-terminus of **7** (Supplementary Figs. 28–30 and Supplementary Table 2). This crosslink (MAbi) is unique in natural products and further highlights the unique structure of **7**. Overall, the combination of these unusual chemical and structural features makes **7** unique among peptides in related families.

The primary structure of **7** (Fig. 3a and Supplementary Fig. 31) confirmed the dehydration of genetically encoded Thr-2, 3, 4, 6, 20, 22, and Ser 8. Subsequent decarboxylated Cys-27 at the C-terminal yields the reactive thioenol, followed by cyclization with Cys-11 and Dhb-24 to generate the *bis*-thioether crosslink. The Aaa-7 residue in **7** is derived from the rearranged dehydration of Ser-7. At the same time, it was shown that three Ser residues (Ser-13, 16 and 18) in the genomic sequence were present in the final structure as Ala residues. It is likely that these L-Ser residues are converted to D-Ala. Such a biochemical

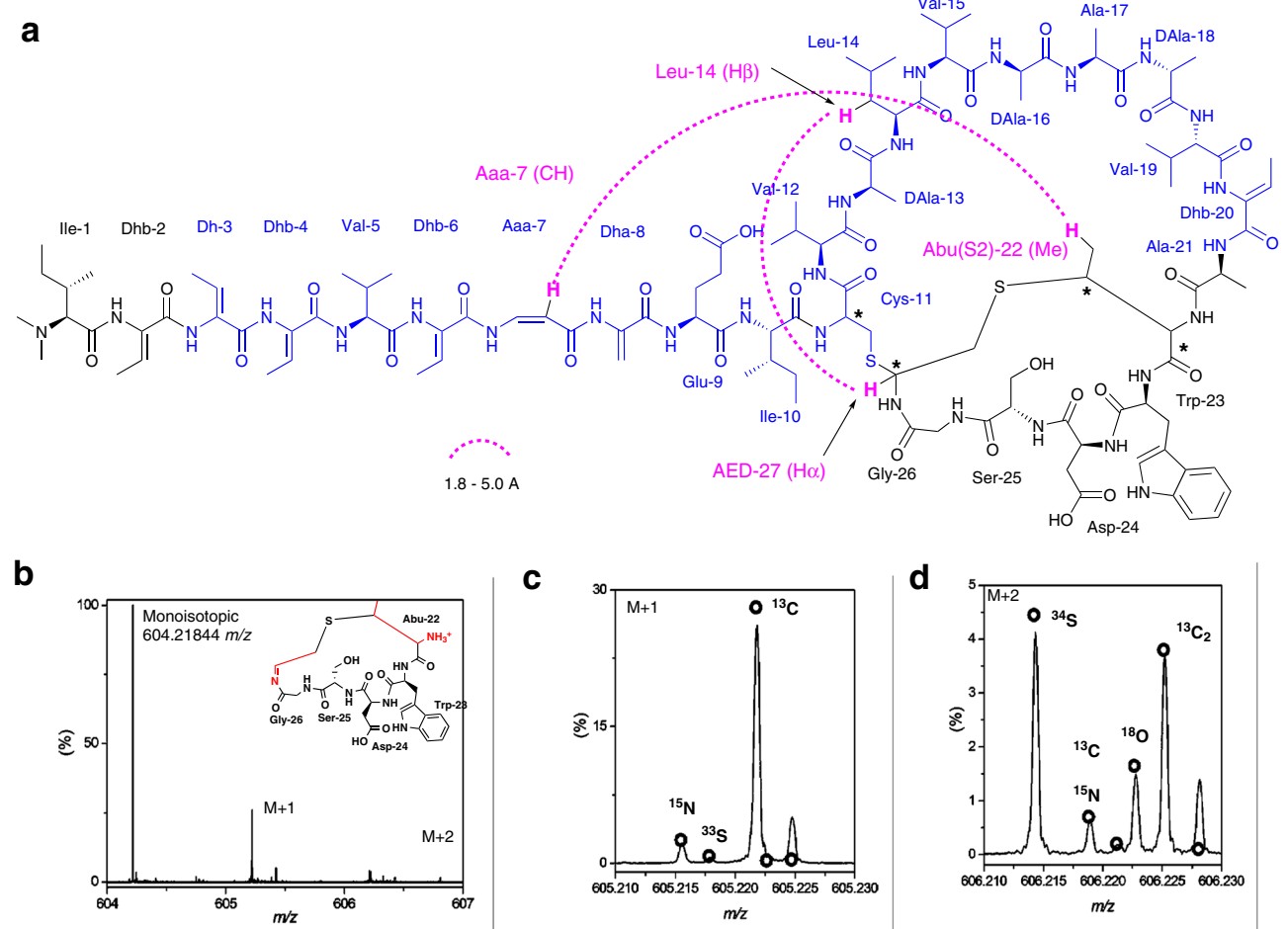

**Fig. 3 | Structure of 7 determined using a combination of NMR, HRMS, genomic and ab-initio molecular dynamics (AIMD)-guided NOE interpretation analyses.** **a** The primary structure of **7**. The residues from Dhb-3 to Ala-21 highlighted in blue were the major fragment which was used for genome mining analysis to identify the *kin* BGC. The configurations shown above were confirmed by either Marfey derivatization or comprehensive NMR elucidation. Two unusual long-range NOE correlations (Leu-14 (Hβ) to AED-27 (Hα) and Aaa-7 (CH)–Abu(S2)-22 (Me)) (pink dash line) were identified through the interpretation of NOE NMR

spectrum. Key local NOE correlations (short and medium ranges) were shown in Supplementary Fig. 31. Stereogenic centres indicated by stars (*) were tentatively assigned to be 11 *R*, 22α*S*, 22β*S*, and 27 *S* based on the comparison of experimental long-range NOE correlations with AIMD calculation. **b** The overall isotopic signal for the y₆ ion to confirm elemental formula. **c, d** The isotopic fine structure of the M + 1 and M + 2 signals are shown in more detail with the theoretical isotope signals for $[C_{26}H_{34}N_7O_8S_1]^+$ ion shown overlaid as a scatter plot (green). Individual isotopologues are annotated.

precedent has been found in a few RiPPs[4,16–18], suggesting that L-Ser undergoes dehydration, followed by subsequent reduction. To confirm those conversions and determine the absolute configurations for all the amino acids in **7**, we performed advanced Marfey's analysis (Supplementary Fig. 32 and Supplementary Table 3). The unmodified amino acid residues are L-configured, except the Cys-11 and Ala residues. Both L- and D-Ala are present in **7** and the relative peak area suggested a molecular ratio of two L-Ala's to three D-Ala's. This is consistent with the formation of D-Ala stereoisomer in **7** exclusively from the three genetically encoded serine residues (Supplementary Fig. 33).

The unusual fused macrocyclic MAbi ring system in **7** marks a structural motif for cyclic peptides with four uncharacterised chiral centres, Cys-11, α- carbon at aminoether 1,2 dithiol-27 (AED-27), and α- and β- carbons at Abu-22 (Fig. 3a). Under the reduced conditions using the previously established method[8], the MAbi residue remains intact, a similar observation also found that the Lan ring in cacaoidin **4** resists reduction[8]. Attempts to identify the derivatives of the crosslinked MAbi residue by advanced Marfey's analysis were unsuccessful, preventing further analysis of using authentic samples to compare any possible synthetic mimics. One possible explanation is that the MAbi ring was degraded to as-yet-unknown fragments under the harsh conditions of Marfey's derivatization.

Interestingly, interpretation of NOE correlations led identification of two unusual long-distance correlations from Aaa-7 (CH) to Abu(S2)−22(Me) and from Leu-14(Hβ) to AED-27(Hα) with estimated distances of 1.8 – 5.0 Å based on previous reports[19,20] (Fig. 3a, Supplementary Fig. 34–36). These long-distance NOE observations only arise from the correct configuration of **7**, which could provide an experimental dataset in comparison with calculated diastereomers generated from computational modelling approaches that have been used in other structurally complex peptides[21–23]. However, due to the high content of modified residues in these systems it is difficult to obtain accurate force fields and so less accurate potentials must be used. As such, we have employed electronic structure calculations at the density functional tight-binding level which avoid the need for force field parameter sets suitable for the compound under study[24]. Distances between atom pairs identified by NMR NOESY correlations (Supplementary Fig. 34–36) were monitored during Ab-initio molecular dynamics (AIMD) simulations using the GFN2-xTB-MD method[24]. Multiple structures containing different combinations of either the *R* or *S* configurations at the chiral carbons in the vicinity of the *bis*-thioether linkage were simulated. Of the structures of candidate isomers studied, three displayed one long-range monitor (Leu-14 (Hβ) to AED-27 (Hα)) satisfying the NOE-predicted distance range of 4–5 Å

(Supplementary Table 4). Only one candidate (stereogenic centres: 11 *R*, 22α*S*, 22β*S*, and 27 *S*) satisfied both long-range monitors (Leu-14 (Hβ) to AED-27 (Hα) and Aaa-7 (CH) to Abu(S2)−22 (Me)), suggesting that this isomer is the only structure capable of generating the observed NOE signals (Fig. 3a, Supplementary Table 5, Supplementary Fig. 37).

Overall, **7** displays unusual chemical features which we named kintamdin. This name is associated with the Kintampo waterfall, the location where *Streptomyces* sp. RK44 was originally isolated. When tested for biological activity, **7** displayed an interesting level of cytotoxicity against cell lines for skin cancer (IC$_{50}$ value of 2.4 ± 0.1 μM), and breast cancer (IC$_{50}$ value of 0.6 ± 0.1 μM), when compared to the control molecule, staurasporine (Supplementary Table 6). However, **7** showed weak or no inhibitory activity against a range of bacterial strains (Supplementary Table 6), indicating that **7** possesses a considerable degree of biological selectivity.

## The minimal BGC of 7

The structure of **7** motivated us to probe its biosynthetic origin in the producing strain, *Streptomyces* sp. RK44. Analysis of the surrounding genetic environment of *kin*A allowed identification of a candidate gene cluster (*kin*) (Fig. 4a and Supplementary Table 7). To validate the identity of the BGC, we carried out a TAR cloning strategy for heterologous expression. To this end, we modified the construction method of pathway-specific capture vectors in order to improve the capture efficiency of the BGC of interest from the genomic DNA of *Streptomyces* sp. RK44 as shown in Supplementary Fig. 38. One out of five clones after yeast transformation was identified to contain the correct length of the BGC in the construct. The construct pCAP03-*kin*2 was then transferred into various *Streptomyces* hosts via *E. coli*-*Streptomyces* conjugation. The production of **7** in *Streptomyces coelicolor* M1152[25] was confirmed through HRMS analysis as well as MS$^2$ fragmentation by comparing with an authentic peptide **7** reference (Fig. 4c, i−ii and xi and Supplementary Figs. 39−40).

In order to determine the boundary of the **7** BGC, a series of gene inactivation were carried out on pCAP03-*kin*2, followed by *E. coli*-*streptomyces* conjugation and fermentation. HRMS analyses of the extracts of these variants demonstrated that gene inactivation of *orf*(−1) and *orf*(−2) as well *orf*1 at the boundaries of the cloned DNA fragment showed no perturbation to the production of **7** (Supplementary Fig. 41). Therefore, the minimal BGC directing the biosynthesis of **7** includes fifteen *orf*s, some of which the functions were assigned (Fig. 4a and Supplementary Table 7).

To assess the in vivo roles of these genes, we generated eight variants (Δ*kin*C-F, H-J and O) (Fig. 4c, traces iii−xi). Gene inactivation of Δ*kin*E resulted in significantly reduced production of **7** whilst knocking out *kin*F caused moderate reduction of the **7** production (Fig. 4c (v−vi)), suggesting the presence of possible synergetic function between KinE and KinF, two putative metallopeptidases, to ensure the efficiency of the proteolytic activities, similar to proposed functions in the biosynthesis of ruminococcin C[26].

The production of **7** was abolished in six other variants (Δ*kin*C, D and H-O), suggesting that these genes are essential for the **7** biosynthesis (Fig. 4c (iii, iv and viii−xi)). No intermediates were accumulated in the four variants of Δ*kin*C, Δ*kin*D, Δ*kin*H and Δ*kin*J (Fig. 4c (iii−iv, viii and x)). KinJ is a putative F$_{420}$H$_2$-dependent reductase that may be responsible for the formation of D-Ala-13, -16 and -18 in **7**, as previously described for LanJ$_B$[16,17] and LxmJ$_C$[4] enzymes. KinC was initially annotated as a hypothetic protein with the length of 323 AA residues. However, structural modelling in Phyre2 Server[27] suggested the C-terminus of KinC (143-229 AA residues) belongs to HopA1 superfamily including phosphothreonine lyases from various Gram-negative pathogens, such as OspF from *Shigella fexneri*[28], which irreversibly removes phosphate groups from phosphothreonine in the activation loop of mitogen-activated protein kinases in the host cells

during the pathogen infections. HopA1-like proteins have been proposed to be responsible for the formation of Dha and Dhb residues in various RiPPs, such as class V lanthipeptides[4,29], thioamitides[7,30–32] and lanthidins[8,9]. Blast search in NCBI suggested that both KinD and KinH are putative phosphotransferases, of which only part of KinD sequences (46-92 AA) share low homology (<34% AA identities) with KinH. Unlike KinD, which is likely to be an active kinase progressing the phosphorylation on Ser/Thr residues, KinH appears to be an inactive phosphotransferase homologue given the absence of the conserved residues in its sequence (Supplementary Fig. 42). Interestingly, recent biochemical investigation demonstrated that the participation of the inactive phosphorylase homologue, TvaE$_{S-87}$, is necessary in the formation of AviCys residue of TVA **2**. Through protein-protein interaction, TvaE$_{S-87}$ coordinates with TvaF$_{S-87}$, a HFCD-like decarboxylase, to form a minimum AviCys synthetase complex for effective AviCys formation[7,31,32]. Similar to this case, it is likely that KinH may coordinate with KinI for decarboxylation on the C-terminal Cys-27 and subsequent cyclization to provide the *bis*-thioether ring system. However, detailed sequence analysis suggests that, while KinI displays low homologues (33% AA identity) with TvaF$_{S-87}$, KinC, KinD and KinH share no significant homologues with the corresponding enzymes in the TVA pathway (Supplementary Table 8). Sequence similarity network (SSN) using the tools of the Enzyme Function Initiative (EFI) with KinC and KinH as query also suggested that both proteins form separate clusters with the corresponding enzymes in other RiPPs (Supplementary Fig. 43). This may not be surprising because both KinCD and KinHI are likely to catalyse different biotransformations, compared to other known RiPPs, to provide the unusual structural elements in kintamdin, such as Aaa and *bis*-thioether macrocycle residues.

Gene inactivation of *kin*O, encoding a putative methyltransferase, resulted in accumulation of a new metabolite **8** in the culture of the Δ*kin*O mutant (Fig. 4c (xi)). MS and MS$^2$ fragmentation analysis demonstrated that **8** is the non-methylated **7** (Supplementary Fig. 44). Knocking out *kin*I, encoding a flavin-dependent decarboxylase, resulted in accumulation of another new metabolite **9** in the culture of Δ*kin*I variant (Fig. 4c (ix)). **9** is the *N*,*N*-dimethylated 27-mer linear peptide containing the intact cysteine residue at the C-terminus and eight dehydrated amino acid residues (1 × Dha, 1 × Aaa and 6 × Dhb) as well as three Ala residues derived from Dha, as evidenced in MS and tandem MS analysis (Supplementary Fig. 45), suggesting that KinI may catalyse the decarboxylation on the C-terminal Cys-27 residue. To this end, overexpression of *kin*O and *kin*I in *E. coli* allowed purification of the recombinant proteins to near homogeneity, as observed in SDS Page analysis (Supplementary Fig. 46). An in vitro assay of a recombinant His$_6$-KinO with **8** in the presence of *S*-adenosyl-L-methionine (SAM, 1 mM) was performed. The production of **7** was confirmed by LC-MS analysis using the authentic sample as comparison (Supplementary Fig. 47), confirming that KinO is responsible for the *N*-terminal Ile dimethylation. An assay of incubating His$_6$-KinI with **9** and other necessary cofactors was performed but no new product was observed, suggesting that **9** is not the bona fide substrate of KinI.

## KinC and KinD catalyse the formation of dehydroamino acid residues in 7

Combined with our heterologous expression, gene disruption and bioinformatic analysis, it was speculated that KinD and KinC may act as a kinase and a phosphoSer/Thr lyase, respectively, on the CP of KinA, to provide dehydroamino acid and Aaa-7 residues in the early stage of the **7** biosynthesis. To this end, attempts to overexpress recombinant proteins, KinC and KinD, as well as fusion proteins of KinC and KinD in *E. coli* were carried out. Unfortunately, no detectable traces of soluble recombinant proteins with or without the inclusion of chaperon proteins were observed, preventing further in vitro assays.

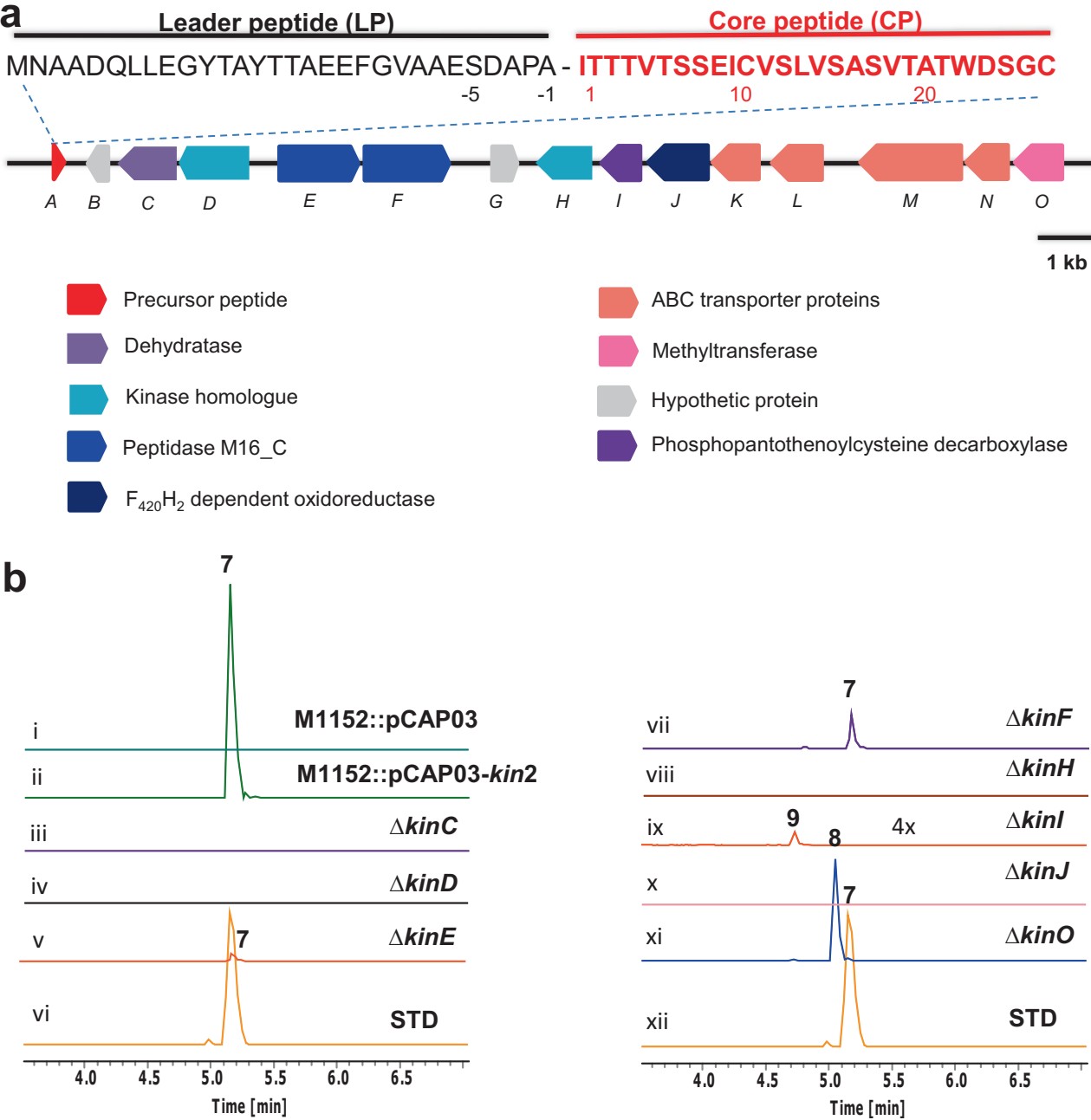

**Fig. 4 | Identification of the minimal BGC of 7 through bioinformatics analysis, heterologous expression and genetic inactivation experiments. a** The minimal BGC of **7** with colour-coded individual genes of proposed functions. **b** Extracted ion chromatogram (EIC) traces of the ions corresponding to peptide **7** and the biosynthetic intermediates, **8** and **9**, in different *Streptomyces coelicolor* M1152 variants. (i). a variant harbouring an empty plasmid, pCAP03; (ii). a variant harbouring the construct with the *kin* BGC; (iii). A knockout variant Δ*kin*C; (iv). A knockout variant Δ*kin*D; (v). A knockout variant Δ*kin*E; (vii). A knockout variant Δ*kin*F; (viii). A knockout variant Δ*kin*H; (ix) A knockout variant Δ*kin*I; (x). A knockout variant Δ*kin*J; (xi). A knockout variant Δ*kin*O; (vi) and (xi). The isolated **7** as the standard.

We developed two co-expression systems of *kin*AD and *kin*ACD with attempts to produce SUMO-fused modified linear precursor peptides, His$_6$-SUMO-KinA(D) and His$_6$-SUMO-KinA(CD), respectively. After purification of the recombinant proteins of interest via immobilised metal affinity chromatography (IMAC), we then adapted the Gluc endoprotease cleavage method[12] to release the leader peptide (LP) and SUMO tag for our MS analysis (Fig. 5a). Indeed, UPLC-QTOF-MS analysis of the Gluc-treated product from co-expression of *kin*AD gave two dominant ions species with *m/z* values of 730.3133 ([M + 2H]$^{2+}$) and 907.9186 ([M + 2H]$^{2+}$), respectively, corresponding to two fragments of the CP. The first fragment has a +80 Da increase, an addition of one phosphate group, compared to the predicted corresponding fragment of unmodified peptide from Ser-(−)5 to Glu-9 (Fig. 5b). Detailed MS$^2$ fragmentation analysis revealed that the phosphate group is located at Thr-2 residue of the CP (Supplementary Fig. 48). The second fragment corresponds to the unmodified 18-mer CP from Ile-10 to Cys-27 that was hydrolysed between Ser-13 and Leu-14 residues, probably due to unspecific hydrolysis, and two cysteine residues (Cys-11 and Cys-27) are oxidised to form a disulphide bridge as evidenced in MS and MS$^2$ fragmentation analyses (Supplementary Fig. 49). Therefore, the corresponding intact compound is likely to be phosphokintamdin, **10** (Fig. 5b), strongly indicating that KinD catalyses the phosphorylation reaction on one amino acid residue in a consecutive fashion at a time starting from the N-terminal Thr-2 residue.

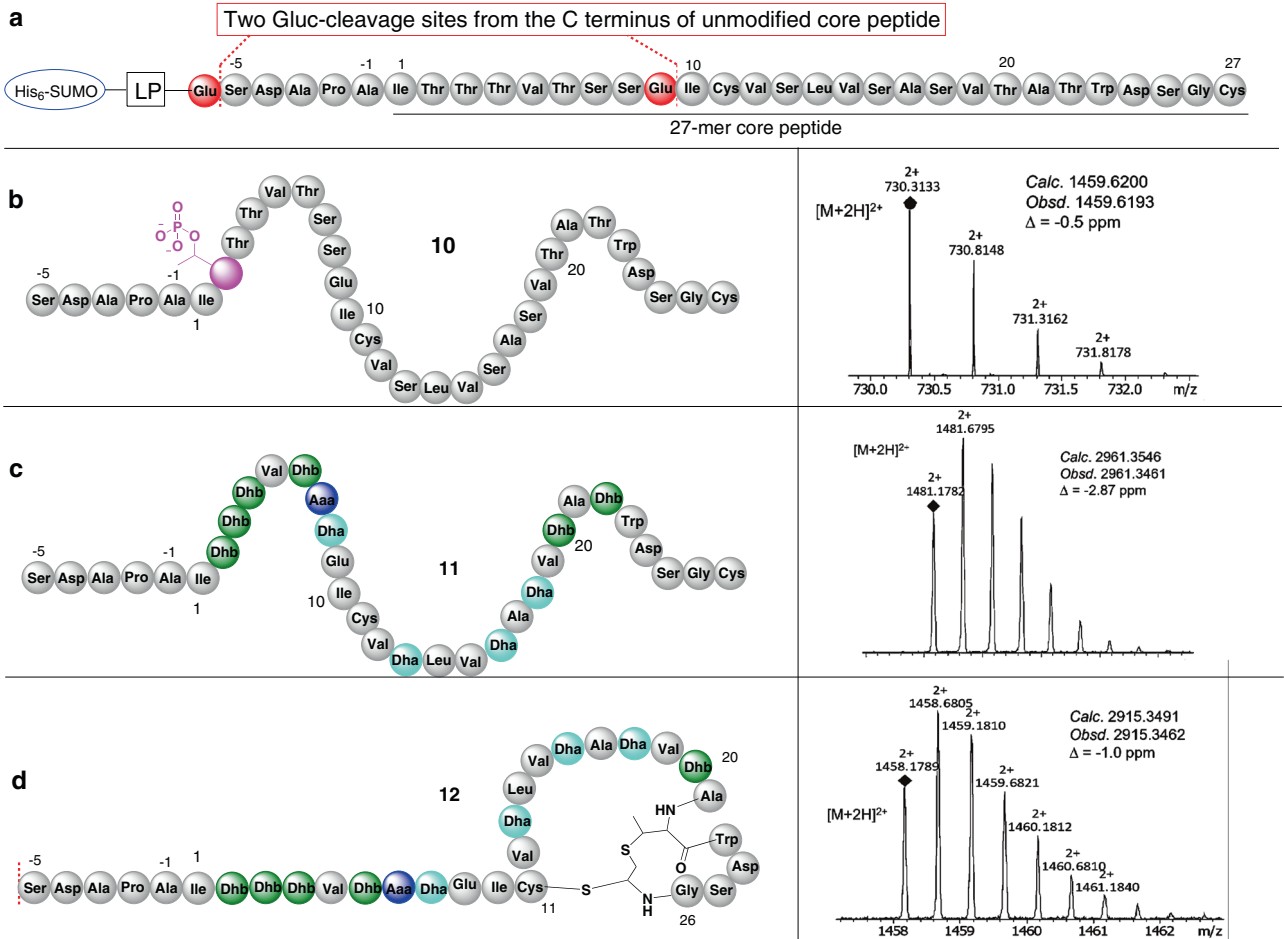

**Fig. 5 | Heterologous expression of genes responsible for the dehydroamino acid and MAbi residues in *E. coli*. a** Schematic structure of the core peptide in SUMO-His₆-KinA. Two cleavage sites (Ser-(-5) and Ile-10) of Gluc protease, highlighted in red, are shown in the C-terminal of the unmodified CP. **b** Schematic structure of the monophosphorylated linear peptide **10** (left) and MS spectrum of the ion (right), corresponding to the monophosphorylated peptidyl fragment (Ser-(−)5 to Glu-9). The genes expressed in *E. coli* include SUMO-His₆-KinA encoding gene and the PTM-encoding gene *kinD*. MS and tandem MS analyses of the other unmodified peptidyl fragment (Ile-10 to Cys-27) was shown in Supplementary Fig. 49. **c** Schematic structure of the fully dehydrated linear peptide **11** (left) and MS spectrum of the ion (right) corresponding to **11**. The genes expressed in *E. coli* include SUMO-His₆-KinA encoding gene and PTM-encoding genes *kinD* and *kinC*. d. Schematic structure of the dehydrated cyclic peptide **12** (left) and MS spectrum of the ion (right) corresponding to **12**. The genes expressed in *E. coli* include SUMO-His₆-KinA encoding gene and PTM-encoding genes *kinD, kin*C, *kin*H and *kin*I. Unmodified AA: grey; targeted residues for PTMs are labelled in colour. Aaa-7: blue, Dhb: green, Dha: light blue; phosphoThr: pink.

In the case of His₆-SUMO-KinA(CD), MS analysis, however, indicated the presence of only one dominant ion with an *m/z* value of 1481.1782 ([M + 2H]²⁺) (Fig. 5c), a −198 Da loss compared to the predicted unmodified CP attached with 5 upstream amino acid residues to Ser-(−)5 with the loss of 11 units of water molecules. Detailed MS² analysis revealed that all of Ser and Thr residues, except Ser-25, were modified to dehydrated amino acid moieties, which are correlated to the pattern of the dehydroamino acid residues in **7** (5 × Dhb residue, 1 Dha, 1 Aaa, 3 × ᴅ-Ala derived from 3 × Dha, 1 × MAbi which consists of one Dhb unit) (Supplementary Fig. 50). As such the corresponding compound was proposed to be the linear dehydrokintamdin **11** (Fig. 5c).

Taken together, our results demonstrated that KinC and KinD are necessary and form a two-component dehydratase in the biosynthesis of **7**.

**The presence of KinI and KinH is necessary for MAbi moiety**
We next carried out the roles of KinI and KinH in the biosynthesis of **7**. Attempts to overexpress *kinH* in *E. coli* was conducted. Unfortunately, no detectable traces of soluble recombinant His₆-KinH with or without the inclusion of chaperon proteins were observed, preventing further

in vitro assays. We then used co-expression systems of *kin*ACDH, *kin*ACDI, and *kin*ACDHI to investigate the roles of KinH and KinI. MS analysis of the Gluc-treated His₆-SUMO-KinA(CDH) from the corresponding co-expression system only gave the ion which is identical to **11** (Supplementary Fig. 51), the sequence of which was confirmed by tandem MS analysis (Supplementary Fig. 51). We also expressed *kin*ACD with *kin*I in *E. coli*, to examine the activity of KinI for Cys-27 oxidative decarboxylation. Analysing the resulting product profile in MS and tandem MS spectra again revealed the only presence of the ion corresponding to **11** (Supplementary Fig. 52), suggesting that, although the presence of the leader peptide, KinI alone is not sufficient for the oxidative decarboxylation and subsequent cyclization. No thioenol or its derivatives, such as aldehyde, alcohol and thiol species, were observed in our case, different to the observations in previous studies of thioamides[7].

Finally, addition of *kin*H into the *kin*CDI *E coli* co-expression system was performed. Analysing the product profile indicated the presence of an ion with an *m/z* value of 1458.1789 ([M + 2H]²⁺) (Fig. 5d). Tandem MS analysis indicated that the corresponding peptide **12** shared the same sequence as kindamdin **7** except the presence of three Dha residues at positions 13, 16 and 18 in **12**, strongly indicating the

presence of MAbi (Supplementary Fig. 53). Clearly, the incorporation of the kinase homologue KinH and the HFCD-like decarboxylase KinI together with KinCD facilitates the MAbi formation, suggesting a coordinated action of KinH and KinI. A similar observation was also reported in the recent investigation of AviCys formation in the TVA biosynthesis that the AviCys formation is dependent on the coordination of the kinase homologue TvaE$_{S-87}$ and the decarboxylase TvaF$_{S-87}$[7,30–32].

### Site-directed mutagenesis to determine key residues in the CP

Interestingly, the N-terminal of the CP is enriched with four Thr (Thr-2, 3, 4 and 6) residues that are converted into corresponding Dhb moieties. It is known that peptides containing dehydroamino acid residues tend to be 2.0$_5$-helix, meaning that the helix is 2 residues per turn and stabilised by H-bonds encompassing 5-membered pseudo-cycles between NH and the carbonyl of the same amino acid residue (Supplementary Fig. 54)[33–35]. As a result, dehydroamino acid residues stretches are arrange in an extended conformation. It was hypothesised that individual Dhb residue or the synergetic impact of these four Dhb residues in the CP could affect the catalytic activity of KinD and/or KinC to provide the Aaa-7 formation. To probe this, a series of alanine scanning mutagenesis was performed, generating six different His$_6$-SUMO-KinA variants, KinA$_{T2A}$, KinA$_{T3A}$, KinA$_{T4A}$, KinA$_{T6A}$, KinA$_{S8A}$ as well as KinA$_{TtoA}$ in which the first four Thr residues (Thr-2, −3, −4 and −6) in KinA, were mutated to Ala residues.

In the case of KinA$_{T2A}$ variant, MS analysis of Gluc-treated His$_6$-SUMO-KinA$_{T2A}$(CD) revealed the presence of an ion with an $m/z$ value of 1475.1887 ([M + 2H]$^{2+}$) (Supplementary Fig. 55), the monoisotopic ion of which has a 12 Da decrease compared to the corresponding ion of **11**. Detailed MS$^2$ analysis revealed that the fragmentation pattern of this ion is identical to **11**, except Ala at 2 position (Supplementary Fig. 55), indicating that Thr-2 is not essential for the formation of Aaa-7 residue (Fig. 6a). The same observations were also found in another four His$_6$-SUMO-KinA variants, KinA$_{T3A}$, KinA$_{T4A}$, KinA$_{T6A}$, and KinA$_{S8A}$, as observed in our MS and tandem MS analyses (Fig. 6a) (Supplementary Figs. 56–59). However, this is not the case of KinA$_{TtoA}$ variant when all of Thr-2, 3, 4 and 6 residues were changed to Ala. Examination of the resulting product profile revealed the presence of two ions with $m/z$ values of 1457.1848 and 1466.1856 ([M + 2H]$^{2+}$) in an estimated ion intensity ratio of 1:1, respectively (Supplementary Fig. 60). The first corresponding peptide has a −18 Da (H$_2$O) difference compared to the latter one. The first peptide was characterised to possess seven fully dehydrated amino acid residues (Fig. 6b and Supplementary Fig. 61). In the case of the latter ion, Ser-7 residue of the corresponding peptide was retained, and the partially dehydrated peptide possess only six dehydrated amino acid residues (Fig. 6b) as observed in MS$^2$ analysis (Supplementary Fig. 62). It is likely that, although still able to process the dehydration of this mutated KinA$_{TtoA}$ variant, KinD displayed less efficiency of phosphorylation on Ser-7 residue, thereby resulting in the accumulation of the partially dehydrated linear peptide with unmodified Ser-7 residue.

The enzyme partner, KinD and KinC, convert Ser-7 into Aaa-7 residue. To probe their substrate promiscuity, Ser-7 residue was mutated to another β-hydroxyl α-amino acid, Thr, to yield the variant, His$_6$-SUMO-KinA$_{S7T}$. Analysing the resulting product profile demonstrated the presence of the KinA$_{S7T}$-related peptide (Supplementary Fig. 63), the monoisotopic ion of which has a +14 Da increase compared to **11**. The peptide was found to be the **11** analogue with methyl-Aaa-7 (Supplementary Fig. 63), indicating that KinC and KinD are capable of processing Thr-7 residue (Fig. 6c).

Likewise, Cys-11 residue in the CP is the key residue for the formation of MAbi. To confirm this, we changed Cys-11 to Ala to provide the variant, His$_6$-SUMO-KinA$_{C11A}$. To our surprise, no ions corresponding to either intact linear dehydrated or cyclic peptides were found. Analysing the resulting product profile, however, revealed two

KinA$_{C11A}$-related peptidyl fragments. The first one was characterised to be the Ala-11 containing mutated C-terminal fragment (Ile-10 to Cys-27 with 5 dehydroamino acid residues (Dha-13, 16, 18 and Dhb-20, 22)), which was believed to arise from the Gluc-cleavage between Glu-9 and Ile-10 of the corresponding dehydrated product (Supplementary Fig. 64). A similar result was observed in our *kin*AD gene co-expression system. The second one was the N-terminal fragment (Ser-(−5) to Dhb-6-NH$_2$ with 4 dehydroamino acid residues (Dhb-2, 3, 4, 6)) (Supplementary Fig. 64), suggesting that the resulting N-terminal fragment (Ser-(−5) to Glu-9) after Gluc treatment was unstable and tends to undergo spontaneous cleavage between Dhb-6-CONH$_2$ and the *cis*-alkene moiety (presumably due to immediate cleavages during the MS analysis). It is likely that substitution of Cys-11 to Ala allowed Gluc protease to access the cleavage site between Glu-9 and Ile-9 even though the CP was modified. Clearly, the intact PTM-processed peptide from this variant is the fully dehydrated peptide (Fig. 6d). Interestingly, no peptidyl fragment containing AviMeCys was found, suggesting that no decarboxylation on Cys-27 occurred in the KinA$_{C11A}$ variant, even in the presence of both KinH and KinI.

Taken together, the above analyses strongly indicated that the synergetic presence of the enriched dehydroamino acid residues in the N-terminal of CP plays an important role in the formation of Aaa-7 residue and KinCD also has capacity of converting Thr-7 residue into methyl-Aaa residue. Changing Cys-11 to Ala completely abolished the cyclization but not the KinCD-catalysed dehydration processing, strongly indicating that Cys-11 is key to the coordinated action by KinH and KinI.

### Prevalence of the *kin* cluster

The structure of **7** and the unique sequences of KinA led us to further address the occurrence of its homologues in bacterial genomes. Searches with KinA as a query in the NCBI database provided the occurrence of its homologues in actinobacteria. Comparisons among these sequences revealed the occurrence of *kin*A-like *orf*s which encode AA sequences with the conserved signature motif of S−S/T−X−X−C−X$_n$−T−X$_4$−C (S7-S8-C11-T22-C27 in the of **7**) (Supplementary Fig. 65). Similar to the CP of KinA, these putative precursor peptides contain enriched Thr residues (3–5) in the corresponding CPs.

The occurrence of the *kin*A-like *orf*s always coincides with the presence, in close proximity, of genes with homology to seven genes in the *kin* BGC, namely *kin*C-F and H-I in *Streptomyces*, and rare actinomycetes (i.e. *Actinopolyspora*, *Nocardioides*, thermophilic *Microbispora*) (Supplementary Fig. 66). While some of these identified BGCs encode putative methyltransferases, others lack the encoded methyltransferases, suggesting the presence of demethylated kintamdin-like RiPPs. One of these BGCs, encoding two putative precursor peptides, was identified in *Streptomyces kurssanovii* NCIMB12788 available in our laboratory (Supplementary Fig. 67). Screening of *S. kurssanovii* in various culture broths resulted in the identification of two ions with $m/z$ values of 1025.1783 and 1009.5079 ([M + 3H]$^{3+}$) as observed in our MS analysis, respectively. The sequences of these peptides confirmed by MS$^2$ fragmentation analysis were correlated to two predicted ones (Supplementary Figs. 68 and 69, respectively), strongly suggesting the presence of two demethylated kintamdin-like peptides.

Given invariant co-occurrence of homologous genes of *kin*C, *kin*D, *kin*H and *kin*I, as well as structural uniqueness among RiPPs, we propose that kintamdin **7** is the founding member of a previously undescribed family of RiPPs, which we named the 'β-bithionins'.

## Discussion

RiPPs are a group of structurally complex naturally occurring metabolites. Although exploring only 20 proteinogenic amino acids, their biosynthetic pathways recruit many PTM enzymes to install diversified structural features in RiPPs including Lan/Lab/Avi cyclic systems that are unique in RiPPs. Kintamdin **7** contains an MAbi crosslink motif,

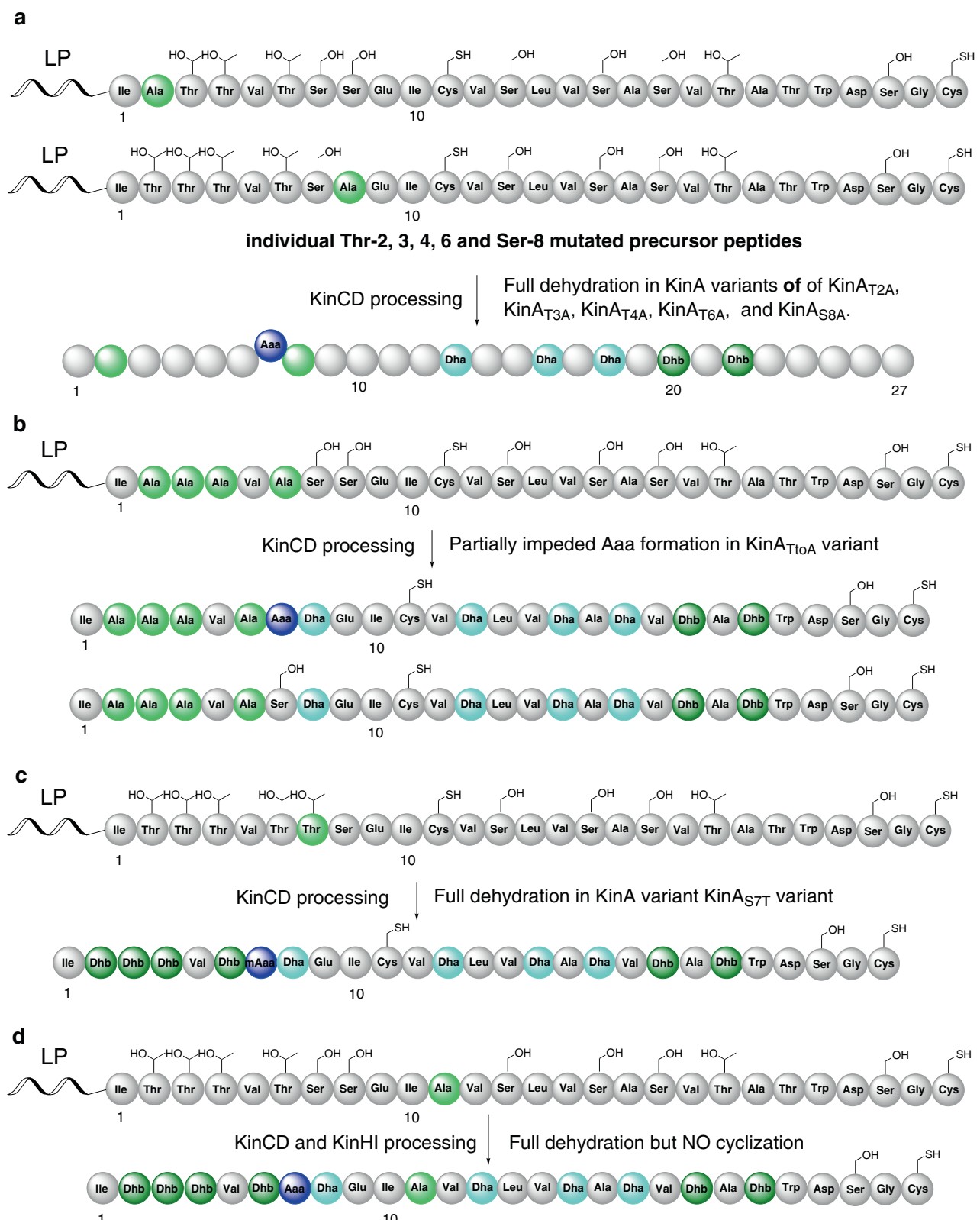

**Fig. 6 | Schematic representation of results from amino-acid substitution experiments in the CP of KinA, designed to probe the key residues in the biosynthesis of 7. a** Individual amino-acid substitution of Thr-2 or -3 or -4 or -6→Ala and Ser-8→Ala had no impact on the dehydration reaction. **b** Mutation of all of the first four Thr residues (Thr-2, 3, 4 and 6) to Ala partially impede Aaa formation. In some cases, Ser-7 residues were retained. **c** Substitution of Ser-7→Thr resulted in identification of fully dehydrated peptide containing methyl-Aaa-7 (mAaa) residue. **d** Substitution of Cys-11→Ala only resulted in the fully dehydrated peptide. No cyclization occurred. The AA residues subjected to site directed mutagenesis were highlighted in light green colour. Dha: light blue; Dhb: deep green; Aaa: blue.

which, to our knowledge, is unique in natural products. Furthermore, an unusual β-enamino acid, (Z)−3-amino-acrylic acid (Aaa/(Δ$^Z$βAla)), was identified at the N-terminal of **7**. β-Amino acid residues were largely thought to be a hallmark of NRP synthesis. Recent biochemical characterisations, however, indicated that β-amino acid residues do occur on peptides synthesized by ribosome[11,12]. The Aaa (Δ$^Z$βAla) residue is rare in the natural product inventory. Only one cyclopeptide metabolite containing the Aaa residue has been discovered from a marine gut fungus, *Aspergillus flavipes*, from *Ligia oceanica* thus far[36]. However, the bio-origin of this cyclopeptide has remained elusive.

Combining structural elucidation, bioinformatics, genetic and biochemical analyses, a biosynthetic pathway of kintamdin **7** is proposed as shown in Fig. 7. KinCD is responsible for Ser/Thr dehydration to install dehydroamino acid moieties in **7**. It is highly likely that the enzyme partner of KinD and KinC is processive in the phosphorylation and elimination of one amino acid residue at a time, starting from *N*-terminus of the CP (Fig. 7). This is not the case for the TVA biosynthesis where the phosphorylation and elimination are likely to initiate from the C-terminal Thr residue in the corresponding CP[7,31,32].

After processing the dehydration on the first four Thr residues in the CP of KinA, the modified CP is likely to be fully extended. Such synergetic conformation changes are likely to play an important role of KinD-catalysed phosphorylation (Fig. 7). Changing all of these four Thr residues to Ala resulted in less phosphorylation efficiency on Ser-7 residue in the proceeding peptide as observed in our mutagenesis experiments, leading to accumulation of the partially dehydrated peptide (Fig. 6c). The stretch proceeding CP is also likely to affect KinC activity, leading to the abstraction of the NH hydrogen of the amide between Dhb-6 and phosphoSer-7 instead of the phosphoSer-7 Cα-H, followed by a nucleophilic attack at the β-carbon of phosphoSer-7 to yield an aziridine intermediate (Fig. 7a). Future structural biology of KinC and KinD will shed light on the underlying mechanisms of KinD-catalysed phosphorylation and KinC-mediated elimination and rearrangement. Interestingly, such activation and aziridination are akin to the Mitsunobu-like reaction used for the conversion of L-Thr residue in peptides into an aziridine motif. In this reaction, triphenylphosphine (Ph$_3$P) combines with diethylazodicarboxylate to form oxyphosphonium intermediate at the β-oxygen of L-Thr, which undergoes an intramolecular cyclization attacked by the deprotonated amide-NH group to generate an aziridine ring (Supplementary Fig. 70)[37,38]. Subsequently, KinC could further catalyse a ring opening on the aziridine-containing intermediate to provide Aaa residue. A similar ring-opening transformation was also found in the conversion of a strained tricyclic aziridino compound into a β-enamino acid system under basic conditions[39]. When Ser-7 was mutated to Thr, KinCD was able to process this residue to produce methyl-Aaa, which is comparable to the chemical activation and aziridination on the Thr residue[37,38] (Supplementary Fig. 70).

After six more rounds of dehydration reactions to provide the fully dehydrated peptide, the kinase homologue KinH coordinates with the HFCD-like decarboxylase KinI to catalyse Cys oxidative decarboxylation and subsequent Michael-type conjugate additions among the resultant thioenol moiety, Dhb22 and Cys-11 to give the MAbi ring system (Fig. 7). The presence of KinH is crucial for the decarboxylation activity of KinI. No compounds containing thioenol moiety or its derivatives were observed in our co-expression system of *kin*ACDI where *kin*H was absent. However, when *kin*H was added into this aforementioned co-expression system, the MAbi moiety was formed in the resulting peptide. Furthermore, the presence of Cys-11 residue in KinA is key to the biosynthesis of **7**. Changing Cys-11 to Ala appeared to have detrimental effects on the activities of KinHI. No cyclic peptide was observed (Fig. 6d).

KinJ, a putative F$_{420}$H$_2$-dependent reductase, could catalyse three stereospecific reductions on the Dha-13, -16 and -18 residues of the resultant cyclic peptide to yield three D-Ala moieties (Fig. 7). No

apparent intermediate was observed in the culture broth of the Δ*kin*J variant, suggesting the instability of the cyclic peptide in the cells of *Streptomyces* heterologous system. However, once D-Ala-13, -16 and -18 residues are formed, the resulting modified precursor peptide containing D-AA residues could confer resistance to cellular protease hydrolysis, similar to the observation in the lantibiotic lacticin 3147[40]. After the bicyclic modified KinA is formed, the leader peptide can be removed by KinEF peptidases to provide **8** (Fig. 4b (xi)), followed by KinO catalyses N-terminal dimethylation to generate the mature kintamdin **7** (Fig. 7). The proteases, KinE and KinF, the reductase KinJ and the methyltransferase KinO display considerable substrate promiscuity. When *kin*I was inactivated to provide the Δ*kin*I variant, three Dha residues (Dha-13, 16 and 18) in the linear dehydrated precursor peptide could be further reduced to the corresponding D-Ala residues. Subsequent removal of the leader peptide and dimethylation provide the shunt intermediate **9** (Fig. 4b (ix)).

In conclusion, we have discovered and characterised kintamdin **7**, a RiPP containing unusual chemical features through a combination of chemical workup, tandem MS analysis and structural elucidation assisted by computational modelling studies. Natural product **7** contains the rare β-enamino acid, (Z)-3-amino-acrylic acid (Aaa or Δ$^Z$βAla), and an unprecedented *bis*-thioether macrocyclic crosslink (MAbi) motif. The minimal *kin* BGC of **7** contains a unique set of biosynthetic enzymes, the kinase KinD, the lyase KinC, the HFCD-like decarboxylase KinI and the kinase homologue KinH, that share low homologues with known enzymes in other RiPP pathways. Gene co-expression in *E. coli* indicated that KinCD enzymes are responsible for introducing dehydroamino acid residues including Aaa-7 residue, and KinI together with KinH catalyses oxidative decarboxylation on the C-terminal Cys-27 and subsequent cyclization to provide the *bis*-thioether MAbi formation. Site directed mutagenesis on KinA demonstrated that the synergetic impact of the first four Thr residues (Thr-2, 3, 4 and 6) in the CP play crucial roles on the formation of Aaa residue. Mutation on Cys-11 to Ala in the CP completely abolished the decarboxylic cyclization reaction. Conserved genomic analysis also allowed identification of the occurrence of two kintamdin-like peptides among the *kin*-like BGCs, which appears to be widespread in actinobacteria.

## Methods

### General chemicals, reagents and analytical methods

[1]H-NMR spectra were obtained on a Bruker AVANCE III HD 400 MHz (AscendTM 9.4 Tesla, UK) with Prodigy TCITM cryoprobe at 298 K in CD$_3$OD and DMSO-$d_6$ (Goss Scientific, Massachusetts, MA, USA). Chemical shifts are reported in parts per million (ppm), relative to the solvent signals. [13]C NMR spectra were obtained with proton decoupling on the same NMR spectrometer and are reported in ppm with TMS for internal standard. Multiplicity is defined as: s = singlet; d = doublet; t = triplet; q = quartet; m = multiplet; br = broad, or combinations of the above. Coupling constants (*J*) are reported in Hertz. High-resolution mass spectra (HRMS) for chemical workup and molecular networking were obtained on either a 12 T SolariX 2XR FT-ICR MS (Bruker Daltonics) via direct infusion or an LTQ Orbitrap Thermo Scientific MS system coupled to a Thermo Instrument HPLC system (Accela PDA detector, Accela PDA autosampler, and Accela pump). The injected samples were chromatographically separated by a C18 (Sunfire 150 × 46 mm) column. The gradient elution for separation was CH$_3$CN/H$_2$O with 0.1% trifluoroacetic acid (TFA) (from 0 to 100% for 30 min, flow rate, 1.0 mL/min, UV detection max 340 nm).

### Structure elucidation of 7

Detail analysis of [1]H, [13]C, COSY, gHSQC, HSQC-TOCSY, gHMBC and ROESY were collected using Bruker Topspin 3 and processed using Mestrenova 12.0 for elucidating the structure of kintamdin (Supplementary Figs. 1–7). [1]H spectra indicated the usual fingerprint of peptides exhibiting multiple exchangeable protons (δ$_H$ 7.6–10.4 ppm),

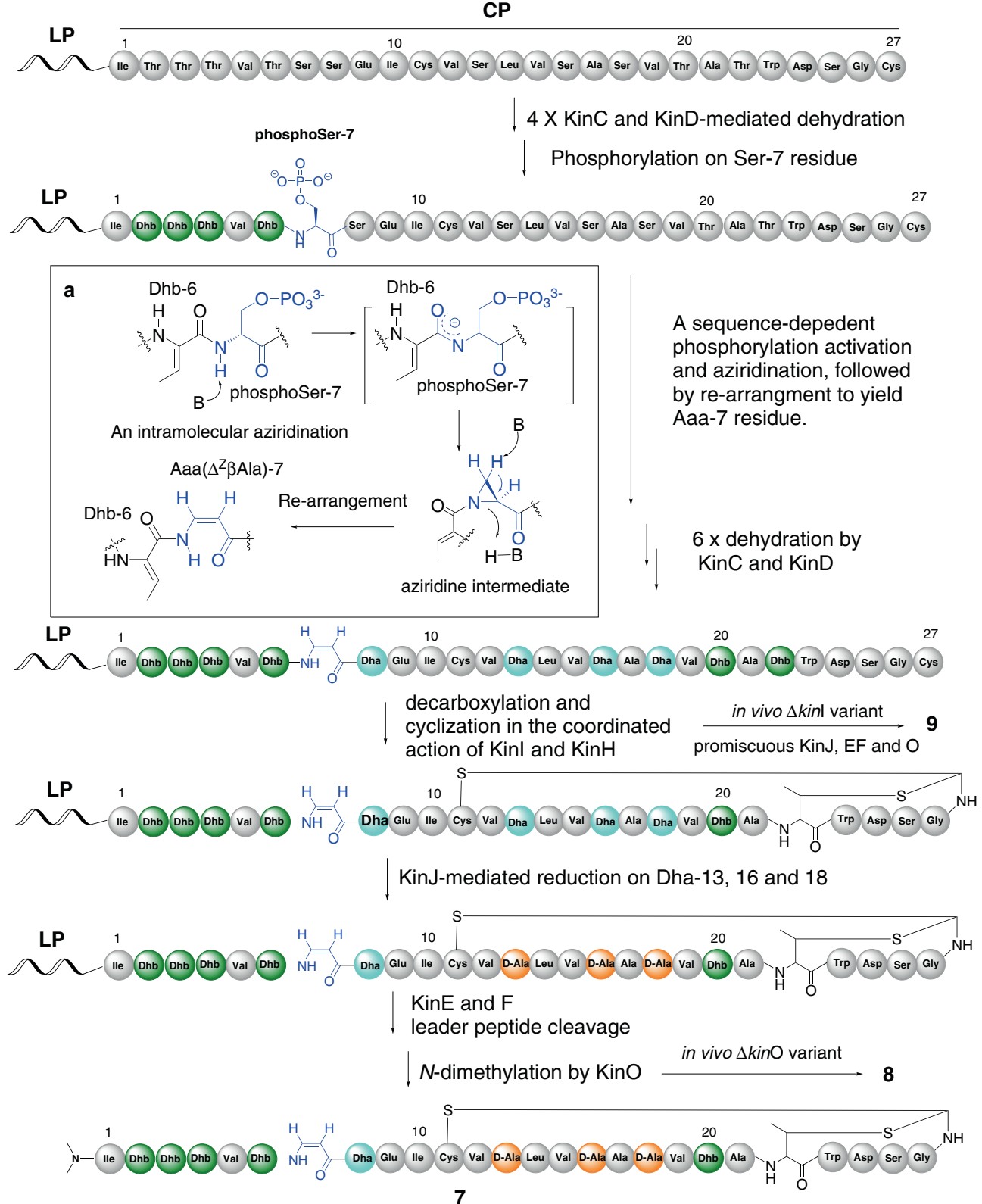

**Fig. 7 | A biosynthetic model of kintamdin 7 with the proposed formation of Aaa-7 (ΔZβAla) (a) and MAbi moieties.** The modified amino-acid residues by PTMs are colour coded. Aaa-7: blue, Dhb: green, Dha: light blue, ᴅ-Ala: orange, MAbi: black. LP leader peptide, CP core peptide.

numerous overlapping α protons of amino acids (δ$_H$ 3.4–4.6 ppm), various alkyls (δ$_H$ 0.8–3.3 ppm), aromatic (δ$_H$ 6.9-7.6 ppm) sidechains. Additionally, there were nine olefinic protons (δ$_H$ 5.3–7.2 ppm) and two chemically equivalent protons (δ$_H$ 2.89 ppm) assigned to a *N,N*-dimethyl group found in the spectrum of **7**. Analysis of COSY, HSQC

and HSQC-TOCSY spectra enabled the assignment of various proteinogenic amino acids such as Trp (×1), Val (×3), Ile (×1), Glu (×1), Asp (×1), Gly (×1), Ser (×1), Ala (×5). Also observed were 7 dehydrated amino acids (DHAAs) such as Dha (×1), and Dhb (×5). It also accounted for one MAbi linkage. Structural configurations of Dhb at positions 2, 3, 4, 6,

and 20 have been deduced as $Z$ based on observed NOE correlation between the Me($\gamma$) group and its corresponding NH (Supplementary Fig. 2).

## FT-ICR mass spectrometry

Peptide samples were ionised by electrospray (nESI) at a typical concentration of $5\,\mu$M. MS analyses were performed on a SolariX FT-ICR instrument equipped with an Infinity ICR cell and a 12 T magnet (Bruker Daltonics, Bremen, Germany). For intact mass analysis, spectra were acquired between m/z 500 and 5000, to yield a broadband 1 or 2 MW time-domain transient. Ion accumulation was set to between 50 and 200 ms, and typically each spectrum was the sum of 50 acquisitions.

Peptide charge state were selected for MS/MS and isolated using CASI (continuous accumulation of selected ions) with a quadrupole width of 2–5 m/z. Fragmentation was performed using both collision-induced dissociation in the collision cell and electron capture dissociation (ECD) using a heated hollow dispenser cathode in the ICR cell. For ECD, typical cathode conditions were bias voltage 1.5 V, lens voltage 15 V, and a pulse length of between 10 and 30 ms. MS2 spectra were recorded between m/z 300 and 5000 and were the sum of 100 1 MW time-domain transients. Pseudo MS3 experiments were performed by first inducing in-source fragmentation by increasing the voltage applied to Funnel 1. A specific fragment ion was isolated using CASI and subject to further fragmentation using either CID or ECD as described above. Data analysis was performed using Bruker DataAnalysis 5.1 (Bruker Daltonics) and, if required, monoisotopic masses were determined using the sophisticated numerical annotation procedure (SNAP; Bruker Daltonics).

## Gene co-expression in *E. coli*

The genes encoding KinA, KinD, KinC, KinH and KinI were synthesized by Genscript and inserted into *E. coli* Duet expression vectors to yield pCDFDuet-*kinA*-*kinD*, pET-*kin*C, pETDuet-*kinC*-*kinI* and pRSFDuet-*kin*H. In the case of generating modified fusion proteins, His$_6$-SUMO-KinA, the *kinA* gene was fused with a small ubiquitin-related modifier (SUMO) gene using In-Fusion® HD Cloning kit with primers Sumo_for and Sumo_Arev (Supplementary Table 11). *E. coli* BL21 (DE3) cells (New England Biolabs, catalogue number: C2527H) harbouring the plasmids or were cultured in LB medium (300 mL) containing streptomycin (50 mg/L) with or without ampicillin (50 mg/L) or kanamycin (50 mg/L) at 37 °C. These co-transformed plasmids include (1) pCDFDuet-*sumo-kinA*-*kinD* for producing KinA(D); (2) pCDFDuet-*sumo-kinA*-*kinD* with pET-*kinC* for producing KinA(CD); (3) pCDFDuet-*sumo-kinA*-*kinD* with pETDuet-*kinC*-*kinI* for producing KinA(CDI); (4) pCDFDuet-*sumo-kinA*-*kinD* with pRSFDuet-*kin*H for producing KinA(CDH); (5) pCDFDuet-*sumo-kinA*-*kinD* with pETDuet-*kinC*-*kinI* *and* pRSFDuet-*kin*H for producing KinA(CDHI).

To the culture IPTG (0.1 mM) was added when the OD$_{600}$ reached 0.6. The induced cultures were further cultured for 20 h at 16 °C. The resultant cultures were collected by centrifugation (5000×g for 20 min at 4 °C) and stored at −80 °C prior to purification.

## Construction of the recombinant strains for producing the KinA variants

To change the KinA residues in the N-terminal CP region to provide seven gene variants, we performed a whole plasmid PCR using two corresponding pairs of overlapping primers and pCDFDuet-*sumo-kinA*-*kinD* as the PCR templates. These include (1) thr2-ala-F and thr2-ala-R for producing *kin*A$_{T2A}$ variant; (2) thr3-ala-F and thr3-ala-R for producing *kin*A$_{T3A}$ variant; (3) thr4-ala-F and thr4-ala-R for producing *kin*A$_{T4A}$ variant; (4) thr6-ala-F and thr6-ala-R for producing *kin*A$_{T6A}$ variant; (5) ser7-thr-F and ser7-thr-R for producing *kin*A$_{S7T}$ variant; (6) ser8-ala-F and ser8-ala-R for producing *kin*A$_{S8A}$ variant; (7) cys-11-ala-F and cys-11-ala-R; hr2-ala-F and thr2-ala-R for producing *kin*A$_{C11A}$ variant.

The corresponding sequences of these primers are listed in Supplementary Table 11. After addition of DpnI (Thermo Scientific) to digest methylated parent plasmid, the resulting plasmids were purified by agarose gel electrophoresis, for expression of the corresponding KinA mutants.

To generate the remaining variant, *kin*A$_{TtoA}$, a nested PCR was performed using two pairs of overlapping primers. These include (1) PCDFmut-f1 and thr2346-ala-r1; thr2346-ala-f2 and PCDFmut-r2 for producing two fragments of *kin*A$_{TtoA}$. The resulting two fragments were purified by agarose gel electrophoresis, respectively, and added together with the linearised plasmid (NcoI and EcoRV (NEB)) for in-fusion ligation, giving the plasmid that express the corresponding KinA$_{TtoA}$ mutant. The plasmids were confirmed by DNA sequencing.

Each plasmid except the one containing *kin*A$_{C11A}$ was co-transferred with pET-*kin*C plasmid into BL21(DE3) competent cells for gene expression as stated above.

The plasmid containing *kin*A$_{C11A}$ was co-transferred with pETDuet-*kinC*-*kinI* and pRSFDuet-*kin*H plasmids into BL21(DE3) competent cells for producing KinA$_{C11A}$(CDHI) variant.

## Purification of modified His$_6$-SUMO-KinA and its variants from *E. coli*

The cell pellets of co-expression cultures were resuspended in buffer A (50 mM Tris-HCl, 0.3 M NaCl, 10 mM imidazole, pH 8.0), followed by lysis with an ultrasonic processor. After centrifugation to remove cell debris in the lysate, (5000 × g, 4 °C for 15 min), the resultant supernatant was filtered through a 0.45 μm filter prior to purification with HIS-Select® Nickel Affinity Gel (Millipore). The resin was washed with 15 mL of buffer A, followed by two times of washing with buffer B (50 mM Tris-HCl, 0.3 M NaCl, 20 mM imidazole, pH 8.0). The recombinant protein was eluted with 10 mL of buffer C containing imidazole (200 mM) and concentrated to 0.5 mL using an Amicon Ultra-15 Centrifugal Filter Unit (Millipore). The purified protein was store at −80 °C in storage buffer (100 mM Tris-HCl, pH 8.0, 150 mM NaCl, 10% (w/v) glycerol, 1 mM DTT).

## Proteolytic cleavage of modified His$_6$-SUMO-KinA or its variants by GluC protease

The modified His$_6$-SUMO-KinAs were diluted to a concentration of 2 mg/mL with 50 mM HEPES buffer (pH 7.5). To the peptide solution (90 μL), GluC protease (Sigma Aldrich) (10 μL, 2 mg/mL) was added. The mixture was incubated for 12–16 h, followed by quenching with trifluoroacetic acid (TFA, 0.1%) prior UPLC-HRMS analysis.

## Reporting summary

Further information on research design is available in the Nature Research Reporting Summary linked to this article.

## Data availability

Data supporting the findings of this work are available within the paper and its Supplementary Information files. A reporting summary for this Article is available as a Supplementary Information file. The datasets generated and analysed during the current study are available from the corresponding author upon request. GenBank accession numbers used are reported in Supplementary Tables 7, 8 and 12.

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

## Acknowledgements

This work received financial support from Biotechnology and Biological Sciences Research Council UK (S.W. and H.D., BB/P00380X/1 and BB/R00479X/1, D.J.C., BB/R013993/1), Scottish Funding Council Covid-19 Grant extension and Bridging Fund (S.W. and H.D.); UKRI Covid-19 Extension Allocation Fund (S.W. and H.D.); The Elphinstone Scholarship of University of Aberdeen (QF), Leverhulme Trust-Royal Society Africa award (AA090088) and the jointly funded UK Medical Research Council UK Department for International Development (MRC/DFID) Concordat agreement African Research Leaders Award (MR/S00520X/1 to K.K. and H.D.), and National Natural Science Foundation of China (31570033, 31811530299, and 31870035 to Y.Y., 31929001 to H.D.), and the Royal Society-NSFC Newton Mobility Grant Award (IEC\NSFC\170617 to H.D. and Y.Y.). B.F.M. thanks the Portuguese Foundation for Science and Technology for financial support (POCI-01-0145-FEDER-032229 and CENTRO-01-0145-FEDER-000014, 2017-2020). H.D. and S.W. thank Professor Brady Moore in the Scripps Institute of Oceanography, University of San Diego, USA, for the gift of pCAP003 plasmid, Dr. Vladimir Larionov in National Cancer Institute, National Institute of Health, Bethesda, USA for the gift of *Sacharomyces cerevisiae* VL6-48N strain, Dr Juan-Pablo Gomez-Escribano and Professor Mervyn Bibb in John Innes Centre, UK for the gift of *Streptomyces coelicolor* M1152, and Samantha Law and Carol Philips in NCIMB Ltd (UK) for the kind gift of *Streptomyces kurssanovii* NCIMB12788.

## Author contributions

S.W., S.L., Z.L., K.W. and Y.G. performed genetic, site-directed mutagenesis and in vitro assays. Y.Y. and H.D. performed genome mining and bioinformatic analyses. J.T., Q.F., D.J.C., R.G., H.D. determined chemical structures. B.F.M. performed computational simulations. S.W., S.L., Q.F., B.F.M., J.T., K.K., D.J.C., L.T. and H.D. analysed the data. B.F.M., K.K., J.T., and H.D. wrote the paper.

## Competing interests

The authors declare no competing interests.
