## [Peer Review File · Nature Communications]

A ribosomally synthesized and post-translationally modified peptide containing a β -enamino acid and a macrocyclic motifEditorial Note: This manuscript has been previously reviewed at another journal that is not operating a transparent peer review scheme. This document only contains reviewer comments and rebuttal letters for versions considered at Nature Communications.

REVIEWER COMMENTS

Reviewer #1 (Remarks to the Author):

This manuscript by Wang et al describes the discovery and structural characterization of a RiPP natural product named kintamdin. This compound has two structural features: a beta-enamino acid motif and a C-terminal bicyclic motif. The authors further demonstrate that kintamdin displays selective cytotoxicity against several cancer cell lines but not bacteria. Preliminary biosynthetic investigation is also conducted to identify a minimal BGC for this compound. This reviewer thinks that this study is very interesting and of high novelty that is suitable for publication in Nat. Commun. However, several major concerns need to be addressed:

1. In terms of the stereochemistry of the Abu-22 residue, which is also mentioned by Reviewer #1, the authors should provide more solid experimental evidence. In a molecule of such complexity, interpretation of NOE data combined with computational modeling is not sufficient. In previous studies of stereochemistry determination of lanthionine crosslinks, researchers often synthesize standard compounds to match the crosslinked amino acids generated by acid-mediated hydrolysis of lanthipeptides. As the structural novelty is the main contribution of this study, this reviewer thinks that a solid structural characterization is necessary.
2. More information regarding the biosynthetic process should be provided. The authors provided data regarding the dehydration of Ser/Thr residues in kintamdin precursor peptide by KinC-KinD. However, this information is well documented in the studies of Tao et al. (Angew. Chem. Int. Ed. 2020, 59,18029) and Liu et al (Cell Chemical Biology 2021, 28, 675–685). The biosynthetic formation of a beta-enamino acid motif and a C-terminal bicyclic motif is not sufficiently investigated or discussed. Regarding the beta-enamino acid motif, the authors hypothesized that the installation of multiple Dhb residues N-terminally to Ser-7 is important for its formation. Could several mutagenesis experiments be done to support this hypothesis? The proposal of an aziridine intermediate formation is interesting. Could a Ser-to-Thr mutation of Ser-7 provides any stable intermediate? Maybe a phosphorylated Thr7 residue due to the increased steric hinderance? Overall, several mechanistic hypothesis could be validated by simple mutagenesis studies, which could further improve the quality of this work.
3. Similarly, could mutagenesis experiments be done to further understand the formation of the C-terminal bicyclic motif and the substrate tolerance of the biosynthetic machinery? These studies could further provide kintamdin analogs and maybe preliminary data on the structure-bioactivity relationship.

Reviewer #2 (Remarks to the Author):

The authors have expanded and improved the scope of the experiments in the revised manuscript. They should be commended for these new efforts; however, this reviewer still has two concerns with the manuscript:

1. The presentation remains a little difficult to follow in some instances. This is even somewhat more of a problem with the presentation and discussion of the new data in the revised manuscript. For example, the longer, and more detail focused, discussion does not help. If anything, the updated presentation makes the manuscript more difficult to follow for a general reader, and therefore, reinforces the notion that the work may be more appropriate for a field specific journal.
2. This reviewer still questions the ultimate novelty of the work in light of the fact that this is not the first report of a 3-amino-acrylic acid, and although this may be a new family of the RiPPs (as defined by the rules established by the RiPPs community) the "newness" of these structures in a classic new metabolite sense is not great.

The updated manuscript is definitely technically improved, and I do not oppose publication in Nat.

Comm from a technical standpoint. However, from a pure novelty (and presentation) standpoint the work may be somewhat more appropriate for a more field specific journal.

Minor concern: At the end of the revised abstract the authors have now chosen to highlight the impact of KinCD on “bio-based functional materials”. This seems a little bit of a stretch at this early stage in their understanding of KinCD. The authors might think about revising this updated sentence.

We would like to thank the reviewers for their suggestions and comments. Below we included a point-by-point responses highlighted in blue to all of the comments raised by each reviewer.

Reviewer #1 (Remarks to the Author):

This manuscript by Wang et al describes the discovery and structural characterization of a RiPP natural product named kintamdin. This compound has two structural features: a beta-enamino acid motif and a C-terminal bicyclic motif. The authors further demonstrate that kintamdin displays selective cytotoxicity against several cancer cell lines but not bacteria. Preliminary biosynthetic investigation is also conducted to identify a minimal BGC for this compound. This reviewer thinks that this study is very interesting and of high novelty that is suitable for publication in *Nat. Commun.* However, several major concerns need to be addressed:

1. In terms of the stereochemistry of the Abu-22 residue, which is also mentioned by Reviewer #1, the authors should provide more solid experimental evidence. In a molecule of such complexity, interpretation of NOE data combined with computational modeling is not sufficient. In previous studies of stereochemistry determination of lanthionine crosslinks, researchers often synthesize standard compounds to match the crosslinked amino acids generated by acid-mediated hydrolysis of lanthipeptides. As the structural novelty is the main contribution of this study, this reviewer thinks that a solid structural characterization is necessary.

We appreciate of the above comments. As mentioned in our manuscript and the previous “authors responses to reviewers”, we identified two long-range NOE correlations from our experimental data (Leu-14 (H β) to AED-27 (H α) and from Aaa-7 (CH) – Abu(S2)-22 (Me)). The existence of such NOE correlations from large separations indicated the paradigm that **only** the correctly-folded 3-dimensional geometrical structure gives such interactions. We also simplified the presentation in Fig 3 by moving all of “local” correlation into Supplementary Fig 31 but only displaying these two long-range NOE correlations to illustrate geometries among these residues.

Among our calculated isomer models, three displayed one long-range monitor (Leu-14 (H β) to AED-27 (H α)) satisfying the NOE-predicted distance range of 4-5 Å. Only one isomer with the 11*R*, 22 α *S*, 22 β *S* and 27*S* configurations would have such two long-range correlations (Leu-14 (H β) to AED-27 (H α) and from Aaa-7 (CH) to Abu(S2)-22 (Me)). Based on this interpretation, we tentatively assigned these stereogenic centres.

We indeed tried several reported experimental methods (*Angew Chemie Int Ed.*, 59, 12654 (2020)) to generate MAbi-containing peptidyl fragments as authentic samples so that we can use them to compare synthetic samples. Such methods were also applied in an attempt of structural elucidation of Lan and AviCys moieties of cacaoidin (*Angew Chemie Int Ed.*, 59, 12654 (2020)). Unfortunately, we did not observe any fragments related to MAbi residue in our analysis, similar to what was observed in cacaoidin (*Angew Chemie Int Ed.*, 59, 12654 (2020)) where the Lan moiety was resistant to reduction and degradation. Without such fragments, we won't be able to perform structural comparison at all even if we synthesize fragments of possible isomers. In the case of cacaoidin, the stereochemistry of Lan was tentatively assigned. It is worth noting that macrocyclic rings of several newly discovered RiPPs, such as the avionin-containing lipolantionin (microvionin, *Nat. Chem. Biol.* 14, 652 (2018)), the Lan-containing cacaoidin (*Angew Chemie Int Ed.*, 59, 12654 (2020)), and the Class VI lanthipeptide, lexapeptide, (*Angew Chemie Int Ed.*, 59, 18029 (2020)) were reported as planar structures. We suspected that the authors in these reports may also meet the similar problems to ours.

In our current revision, we have significantly shifted our focus and expand our investigation on the biosynthetic origin of kintamdin, particularly in the formation of Aaa-7 and MAbi ring residues although structural elucidation still plays an important part in the beginning of our report. We also highly appreciate of two reviewers' comments on the assignments of the carbon centres. Therefore, we have moved the structural modelling section from the main text to supplementary information and reported these four carbon centres as tentative assignments, similar to recent reports of RiPPs (*Angew Chemie Int Ed.*, 59, 12654 (2020), and 59, 18029 (2020) and more recently cyclopeptides (*Proc Natl Acad Sci U S A.* **119**, e2117941119 (2022)). Total synthesis of kintamdin isomers will be our next investigation to ascertain the absolute structure with our authentic sample.

2. More information regarding the biosynthetic process should be provided. The authors provided data regarding the dehydration of Ser/Thr residues in kintamdin precursor peptide by KinC-KinD. However, this information is well documented in the studies of Tao et al. (*Angew. Chem. Int. Ed.* 2020, 59,18029) and Liu et al (*Cell Chemical Biology* 2021, 28, 675–685). The biosynthetic formation of a beta-enamino acid motif and a C-terminal bicyclic motif is not sufficiently investigated or discussed. Regarding the beta-enamino acid motif, the authors hypothesized that the installation of multiple Dhb residues N-terminally to Ser-7 is important for its formation. Could several mutagenesis experiments be done to support this hypothesis? The proposal of an aziridine intermediate formation is interesting. Could a Ser-to-Thr mutation of Ser-7 provides any stable intermediate? Maybe a phosphorylated Thr7 residue due to the increased steric hinderance? Overall, several mechanistic hypothesis could be validated by simple mutagenesis studies, which could further improve the quality of this work.

We highly appreciate of the above comments and suggestions which helped us to improve our studies. We performed a series of SDM experiments. Individual Ala scanning SDM generated five His₆-SUMO-KinA variants, KinA_{T2A}, KinA_{T3A}, KinA_{T4A}, KinA_{T6A}, and KinA_{S8A}. MS and tandem MS fragmentation analysis indicated the individual mutagenesis on Thr residue had no impact on the formation of Aaa-7 as well as other dehydroamino acid residues. However, when all of these four Thr residues, namely Thr-2, 4, 5, 6 prior Ser-7, were mutated to Ala to generate the variant His₆-SUMO-KinA_{T10A}, we observed two ions with m/z values of 1457.1875 ([M+2H]²⁺) and 1466.1947 ([M+2H]²⁺), respectively. The first ion corresponds to the mutated core peptide (CP) with seven dehydroamino acid residues (the first four were mutated to Ala) as expected, the sequence of which was assigned by MS² fragmentation analysis. The latter ion is the mutated core peptide with, however, only six dehydroamino acids residues while Ser-7 residue was retained as indicated in our MS² fragmentation analysis. This result indicated that the first four Thr residues are important for the formation of Aaa-7. One possible explanation of this observation is that KinD is a sequence-specific enzyme. Changing all four Thr in the N-terminal of CP resulted in less efficiency of phosphorylation on Ser-7 residue by KinD. As a result, some populations of partially dehydrated CP with Ser-7 retained were observed.

Interestingly, it is well known that peptides containing dehydroamino acid residues tend to be 2.0₅-helix, meaning that the helix is 2 residue per turn and stabilized by H-bonds encompassing 5-membered pseudo-cycles between NH and the carbonyl of the same amino acid residue as shown below (*J. Am. Chem. Soc.* **121**, 3272 (1999), *ChemPlusChem*, **86**, 723 (2021)). As such, dehydroamino acid stretches arrange as flat peptides.

H-bonding pattern in an 3_{10} helix containing normal amino acid residues. 3_{10} helices constitute nearly 10-15% of all helices in protein secondary structures.

H-bonding pattern in a 2.0_5 -helix (fully extended conformation) containing dehydroamino acid residues

Dash lines represent intramolecular H-bonding

It is likely that such fully extended feature in the Dhb-enriched AA sequence prior Ser-7 residue in the CP of KinA not only affect the phosphorylation efficiency of KinD but also allow NH in the amide bond between Dhb-6 and phosphoSer-7 generated by KinD to become more accessible than C α -H of phosphoSer-7 by KinC-mediated abstraction, followed by an intramolecular nucleophilic attack to yield an aziridine intermediate. Future structural biology will shed light on the exact mechanisms of how the substrate is handled by both KinC and KinD.

In a separate case, when Ser-7 was mutated to Thr to provide a variant SUMO-KinA_{STT}, we observed an ion with an m/z value of 1481.1887 ($[M+2H]^{2+}$), the monoisotopic ion of which has a + 14 Da increase compared to **11**. Detailed MS² analysis revealed that the fragmentation pattern of this ion is identical to **11**, except the addition of a methyl group at 7 position (Supplementary Fig. 47), indicating that KinCD is capable of processing Thr residue to generate the methyl dehydrokintamdin (Fig. 6c). It may not be surprising because such a process is comparable to the chemical conversion of L-Thr residue in peptides using Mitsunobu reagents into aziridine motifs. Such a chemical transformation undergoes an intramolecular N-alkylation reaction. It is likely that KinC-mediated aziridine follows the similar chemical pathway as shown below. The facile *anti*-elimination of aziridine intermediate will provide methyl-Aaa residue.

It should be noted that KinC and KinD shares no or low homologues (**Supplementary Table 7**) to the HopA1-like lyases TvaD_{S-87} and the Ser/Thr kinase TvaC_{S-87} in the biosynthesis of thioviridamides. Sequence similarity network (SSN) using the tools of the Enzyme Function Initiative (EFI) with KinC as query also suggested that KinC forms a separate cluster with the corresponding enzymes in the pathways of other known RiPPs. This may not be surprising

because KinCD enzyme partner is likely to catalyse different biotransformation, compared to other known RiPPs, to provide the unusual Aaa structural element in kintamdin.

3. Similarly, could mutagenesis experiments be done to further understand the formation of the C-terminal bicyclic motif and the substrate tolerance of the biosynthetic machinery? These studies could further provide kintamdin analogs and maybe preliminary data on the structure-bioactivity relationship.

We appreciate of the above comments. We indeed performed further investigation of the C-terminal bicyclic MAbi motif by developing three co-expression systems of *kinACDH*, *kinACDI* and *kinACDHI*. Based on our bioinformatic analysis (Supplementary figure 44) and gene co-expression in *E coli*, we concluded that:

1 Bioinformatic analysis suggested that KinH is likely to be an inactive phosphorylase, similar to TvaE_{S-87} in the biosynthesis of thioviridamides. Individual addition of either *kinH* or *kinI* into *kinACD* co-expression system gave the same compound as the one found in *kinACD*. Furthermore, no thioenol species or its derivatives, such as aldehyde, alcohol or thiol in the gene co-expression of *kinACDI*, were found, different to what was observed in the similar reactions in various studies in thioamides (Cell Chem Biol, **28**, 675 (2021); *J Am Chem Soc.* **144**, 5136 (2022); *Org Lett.* **24**, 1518 (2022)). This indicated that, unlike the corresponding HFCD-like decarboxylases in the pathways of various thioamide RiPPs, KinI alone is not sufficient enough to catalyse the oxidative decarboxylation reaction. This may not be surprising because KinI enzyme and/or its partner catalyse a rather different biotransformation, compared to TVA, to provide the unusual *bis*-thioether macrocyclic structural element in kintamdin.

2. Addition of both *kinI* and *kinH* into *kinACD* co-expression system provided the MAbi moiety as evidenced in our MS and tandem MS analyses, indicating that the presence of KinH is essential for the cyclization. It is highly like that the oxidative decarboxylation and subsequent cyclization rely on the coordinated action of KinH and KinI, similar to what was observed in the thioamide system (Cell Chem Biol, **28**, 675 (2021); *J Am Chem Soc.* **144**, 5136 (2022); *Org Lett.* **24**, 1518 (2022)). Despite of considerably structural differences between kintamdin and thioamides, both systems (may extend to cacaoidin, lexapeptide and cypermycin) may adapt a similar coordination action by HFCD-like decarboxylases and inactive kinases with different chemical outcomes (kintamdin for MAbi and thioamides for AviCys).

3. Cys11 is the key residue of MAbi moiety. Mutation of Cys-11 to Ala provided a variant His₆-SUMO-KinA_{C11A}. To our surprise, we didn't observe any traces of intact phosphorylated or dehydrated or cyclised CPs in this system. We were only able to identify two peptidyl fragments which were characterized to be the C-terminal dehydrated peptide (Ile-10 to Cys-27) (Dha-13, 16, 18 and Dhb-22, 24) and a derivative (Ser-(-5) to Dhb-6-NH₂) of the N-terminal dehydrated peptide (Ser-(-5) to Glu-9) (Dhb-2, 3, 4, 6, Aaa-7 and Dha-8). Apparently, substitution of Cys11 to Ala allow Gluc protease to hydrolyse the intact dehydrated CP between Glu-9 and Ile-10. This suggested that Cys-11 residue is important for the KinHI related PTM processing. Without Cys11, KinH and KinI is unable to process the corresponding dehydrated CP.

Taken together, our new experimental data and bioinformatic analysis provided a new line of evidence that the coordinated action of KinI and KinH is responsible for the formation of MAbi moiety. Cys11 residue in the CP play an important role in the biosynthesis, presumably for the enzyme processing of KinHI. Mutating this residue to Ala resulted in full dehydration (11 dehydrated amino acid residues) but no decarboxylation and cyclization occurred.

The focus of the current study is to report the discovery and biosynthetic investigation of this new RiPP. It is our intention to further investigate this system with the aim of generating more bioactive analogues via biotransformation for our future Structural Activity Relationship studies. Our future studies will also include finding a suitable cleavage system to completely remove the leader peptide of KinA after gene expression and purification.

Reviewer #2 (Remarks to the Author):

The authors have expanded and improved the scope of the experiments in the revised manuscript. They should be commended for these new efforts; however, this reviewer still has two concerns with the manuscript:

1. The presentation remains a little difficult to follow in some instances. This is even somewhat more of a problem with the presentation and discussion of the new data in the revised manuscript. For example, the longer, and more detail focused, discussion does not help. If anything, the updated presentation makes the manuscript more difficult to follow for a general reader, and therefore, reinforces the notion that the work may be more appropriate for a field specific journal.

We highly appreciate of the above comments for our improvement. We have made a significant change in our revised manuscript.

Firstly, a detailed introduction was provided as shown in Fig 1, describing the current knowledge of the formation of cyclic ring systems and β -amino acid residues in RiPPs. This added paragraphs highlighted the differences between kintamdin and other known RiPPs and relevant biochemistry as shown in Fig 2.

To avoid a crowded structural presentation in Fig 3, we moved all "local" NOE correlations in the planar structure of kintamdin to Supplementary Information but only showed two identified long-range NOE correlations to support our calculated isomer. In the meantime, we moved the modelling investigation into supplementary information. We balanced our main focus to biosynthetic investigation on kintamdin.

We performed investigation on the biosynthetic origin of both dehydrated amino acid and bis-thioether ring (MAbi) residues (Fig 5). A series of site directed mutagenesis experiments provided a new line of evidence to determine the key residues that play important roles in the formation of Aaa-7 and bis-thioether residues as suggested by the other reviewers (Fig 6). Finally, we reduced the technical discussion, particularly toning down Aaa residue and focused on the proposed biosynthetic model.

Overall, we sincerely hope that our changes would make readers easier to follow.

2. This reviewer still questions the ultimate novelty of the work in light of the fact that this is not the first report of a 3-amino-acrylic acid, and although this may be a new family of the RiPPs (as defined by the rules established by the RiPPs community) the "newness" of these structures in a classic new metabolite sense is not great.

We appreciate of the above comments. Based on the reviewers' comments, we have made significant changes on the narratives in our revision. Although structural elements still play an important part in the beginning of our revision, the majority of our investigation focuses on the biosynthetic origins of Aaa and bis-thioether ring moieties and discussion on mechanistic aspects of these two rare amino acid residues.

The updated manuscript is definitely technically improved, and I do not oppose publication in Nat. Comm from a technical standpoint. However, from a pure novelty (and presentation) standpoint the work may be somewhat more appropriate for a more field specific journal.

We highly appreciate of the above supportive comments. As stated above, we have made changes in presentation and narratives in our revision. We also made significant development to understand the biosynthetic origins of this metabolite. We believed that KinCD and KinHI present two unusual enzyme partners to catalyse the formation of Aaa and bis-thioether ring systems, significantly different to any of other enzymes in RiPPs and other natural products. The topic of the current revision covers the balanced chemistry and biology. We sincerely hope that our changes would make this version suitable for Nat. Comm.

Minor concern: At the end of the revised abstract the authors have now chosen to highlight the impact of KinCD on “bio-based functional materials”. This seems a little bit of a stretch at this early stage in their understanding of KinCD. The authors might think about revising this updated sentence.

We appreciate of the above comments and agree with this reviewer that such a statement is not appropriate at this stage. We removed this statement and remove any implications in bio-based functional materials in our current revision.

REVIEWERS' COMMENTS

Reviewer #1 (Remarks to the Author):

The authors have done a thorough job of responding to my comments as well as the other reviewer's. The additional biochemical experiments and discussions further improve this study of novelty. This reviewer supports its publication in the current form.

Reviewer #2 (Remarks to the Author):

The authors have added a significant quantity of new data to further develop their biosynthetic hypotheses. These data, together with the new text edits and the data included in the last revision, present the manuscript in a "more novel" and improved direction compared to the original submission. As stated before, this reviewer does not object to the publication of the revised manuscript in Nat. Comm. However, from a purely scientific interest standpoint the work may be somewhat more appropriate for a more field specific journal.